# COMBATING DATA LAUNDERING IN LLM TRAINING

## ABSTRACT

Data rights owners can detect unauthorized data use in *large language model* (LLM) training by querying with proprietary samples. Often, superior performance (e.g., higher confidence or lower loss) on a sample relative to the untrained data implies it was part of the training corpus, as LLMs tend to perform better on data they have seen during training. However, this detection becomes fragile under *data laundering*, a practice of transforming the stylistic form of proprietary data, while preserving critical information to obfuscate data provenance. When an LLM is trained *exclusively* on such laundered variants, it no longer performs better on originals, erasing the signals that standard detections rely on. We counter this by inferring the unknown *laundering transformation* from black-box access to the *target* LLM and, via an *auxiliary* LLM, synthesizing queries that mimic the laundered data, even if rights owners have only the originals. As the search space of finding true laundering transformations is infinite, we abstract such a process into a high-level transformation *goal* (e.g., "lyrical rewriting") and concrete *details* (e.g., "with vivid imagery"), and introduce *synthesis data reversion* (**SDR**) that instantiates this abstraction. **SDR** first identifies the most probable *goal* that synthesis should step into to narrow the search; it then iteratively refines *details*, such that synthesized queries gradually elicit stronger detection signals from *target* LLM. Evaluated on the MIMIR benchmark against diverse laundering practices and *target* LLM families (Pythia, Llama2, and Falcon), **SDR** consistently strengthens data misuse detection, providing a practical countermeasure to data laundering.

## 1 INTRODUCTION

*Large language models* (LLMs) now generate text with human-level fluency and stylistic diversity, driving adoption in medicine (Liu et al., 2025), education (Yan et al., 2024), and other high-stakes applications. Such remarkable capabilities demand training LLMs on large-scale high-quality corpora (Wang et al., 2025), whose collection and use, however, are often constrained by privacy and copyright (Li et al., 2023b). A pressing compliance question is whether a deployed LLM was trained on copyrighted or sensitive material without authorization. In post-hoc *unauthorized training data detections*, a data rights owner queries the *target* LLM with proprietary "candidate" texts and compares a per-sample score, e.g., loss (Zhang et al., 2024) or calibrated confidence proxy (Xie et al., 2024), against the score distribution over a held-out non-training texts corpus, following Carlini et al. (2022). The memorization effect of LLMs (Li et al., 2025) implies that training samples tend to receive lower loss or higher confidence, such that a statistically significant score gap indicates the queried sample likely influences training (Figure 1 (a)); mainstream detection methods perform reliably in this "query with originals" regime (see Table 1 "Orig." columns).

This regime presumes that the *target* LLM is always trained on the rights owner's proprietary texts in their *original form*. In practice, however, natural language is malleable; core information and semantics can still be preserved under extensive stylistic and structural transformations (Barzilay & McKeown, 2001; Bhagat & Hovy, 2013) through human writing, programmatic paraphrase, back-translation (Dolan & Brockett, 2005; Bannard & Callison-Burch, 2005), or more recent LLM-enabled large-scale synthesis (Witteveen & Andrews, 2019; Liu et al., 2024b). When an LLM is trained *exclusively* on such transformed surrogates, it does *not* memorize original data and, no longer exhibits a reliable performance gap when queried with the originals (Figure 1 (b)), which erases the signals that unauthorized data detections rely on. Empirically, consider a target Llama-2 (Touvron et al., 2023) trained on a corpus stylistically transformed from Wikipedia articles into lyrics, we find

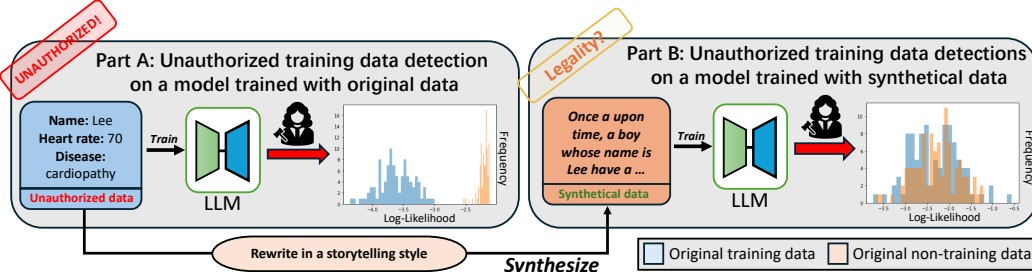

Figure 1: Illustration of how data laundering undermines existing unauthorized training data detections. When unauthorized data is directly used for training, LLMs tend to memorize the unauthorized training data. Training samples exhibit lower loss than non-training data, as shown in Part A. The log-likelihood distributions of training and non-training samples diverge clearly, enabling identification. However, when trained on laundered unauthorized data; as shown in Part B, the distributions of the unauthorized data and non-training samples no longer diverge, preventing reliable identification.

that mainstream unauthorized data detection methods, when tested on the originals (i.e., Wikipedia articles), perform no better than random chance (Table 1 "Syn." columns).

This fragility enables *data laundering*: deliberate obfuscation of data provenance through semantic-preserving transformation to conceal large-scale unauthorized use. Model providers can transform entire proprietary corpora, e.g., synthetically alter personal health records into children's-story-style narratives that retain substantive content, then train on the derivatives, and assert that the original records never entered the training, although sensitive information may still influence the model and can be further leaked. In realistic detections, data rights owners typically have *black-box* access to deployed *target* LLMs via an inference-only API; providers rarely disclose their preprocessing or sourcing pipelines, nor training artifacts. This opacity, where the potential laundering procedure is *unknown* to rights owners, creates a risk to intellectual property and privacy, motivating our question: *How can a data rights owner detect unauthorized data use from black-box model access when the data has been laundered through an unknown transformation?*

Effective detection requires queries that stylistically and structurally resemble what the *target* LLM observed during training. When the laundering transformation is unknown, crafting effective queries becomes an unbounded search over possible data alterations. Our key idea here is to shift the problem from locating specific laundered samples to inferring the *laundering transformation* itself (formalized in Section 3). We make this problem tractable by modeling the unknown transformation as a synthesis procedure defined by a two-level *goal-details* abstraction. A high-level *goal* that captures the primary language register shift[1] (e.g., "rewrite into lyrics") and concrete *details* that refine stylistic and formatting constraints further (e.g., imagery, voice, and rhyme density). Leveraging modern LLMs as controllable generators (Liang et al., 2024a), we instantiate this abstraction as a language-*prompted* specification executed by an *auxiliary* LLM to synthesize candidate surrogates under explicit controls. This *goal-details* schema is compatible with commonly used prompt templates (Mao et al., 2025), allows *goal* to set the coarse-grained stylistic and structural transformation, while *details* provide fine-grained, data-driven synthesis refinement[2].

In Section 4, we introduce *synthesis data reversion* (**SDR**), a two-stage search that returns (i) a *goal-details* synthesis specification and (ii) a set of "training-like" queries synthesized from proprietary texts under detection (i.e., candidate set) and compared against a reference non-training texts (i.e., held-out set), enabling off-the-shelf detection methods, e.g., (Xie et al., 2024) on a *target* LLM.

Mirroring the schema, stage 1 determines the most likely laundering *goal* by screening an established taxonomy of 23 registers (Myntti et al., 2025). For each register, we pre-define a standard rewriting prompt and, with the *auxiliary* LLM, produce short rewrites of a small seed of candidate texts; from these samples we extract a common opening template that captures how the register typically begins. We then task the *target* LLM to score register-conditioned rewrites and keep the few registers that

---

[1]A register is a situational variety of language shaped by purpose, audience, and medium; examples include news, academic prose, instructions, and lyrics (Agha, 2004).

[2]We do not claim true laundering always follows this schema; it is adopted only as a search strategy for synthesizing queries in "training-like" style, which will be used for detection.

Table 1: Performance of unauthorized training data detection on Llama-2 (Touvron et al., 2023) models fine-tuned with either the original MIMIR-wiki (Deng et al., 2023) dataset (Orig.) or its laundered version (Syn.) generated by GPT-4o (Hurst et al., 2024) using the prompt "rewrite in a lyrical style, ensuring the imagery is vivid". Evaluation metrics are defined in Section 5.

| Methods | AUC | | ASR | | TPR@5% | |
|---|---|---|---|---|---|---|
| | Orig. | Syn. | Orig. | Syn. | Orig. | Syn. |
| Loss (Yeom et al., 2018) | 1.000 | 0.539 | 1.000 | 0.565 | 1.000 | 0.040 |
| Ref (Carlini et al., 2022) | 0.971 | 0.603 | 0.920 | 0.610 | 0.850 | 0.100 |
| Zlib (Carlini et al., 2021) | 1.000 | 0.521 | 1.000 | 0.535 | 1.000 | 0.080 |
| Min-K (Shi et al., 2023) | 1.000 | 0.563 | 1.000 | 0.575 | 1.000 | 0.040 |
| Recall(Xie et al., 2024) | 0.999 | 0.558 | 0.995 | 0.565 | 1.000 | 0.000 |

best match the *target* LLM's preferences. Lastly, for each shortlisted register, we synthesize full rewrites of the candidate and reference texts, run unauthorized-use detection, and select the register that maximizes the detection metrics (Algorithm 1). This yields the initial *goal* specification that stage 2 will refine with *details*. Starting from the selected *goal* and its standard prompt, in stage 2 we iteratively infer the missing fine-grained details that make rewrites resemble what the *target* LLM likely saw during training. In each iteration, we sample a seed of proprietary texts; the *auxiliary* LLM rewrites each under the current specification, and *target* LLM generates the next span following the rewrite's opening sentence. The *auxiliary* LLM then summarizes the differences between pairs of rewrites and target-generated follow-ons into refinements to the current specification. Upon using the revised specification to synthesize rewrites for both candidate and reference sets, we accept the revision only if it improves the unauthorized-use detection performance. The loop repeats until the gains plateau or the maximum iteration is reached, yielding a *goal-details* specification and "training-like" surrogates usable with off-the-shelf detections under laundering (Algorithm 2).

In Section 5, we evaluate **SDR** on MIMIR benchmark (Deng et al., 2023) across *target* LLM families (Pythia (Biderman et al., 2023), Llama-2 (Touvron et al., 2023), falcon (Zhang et al., 2022)), and *auxiliary*-LLM choices (DeepSeek (Liu et al., 2024a), GPT-4o (Roumeliotis & Tselikas, 2023), Claude (Wu et al., 2023)) under diverse simulated large-scale laundering procedures. **SDR** consistently strengthens off-the-shelf standard detection methods in *all* detection metrics with ablation studies showing that both stages contribute to the gains.

**Contributions.** This study presents (i) a data laundering-aware, post-hoc unauthorized data detection formulation for black-box LLMs. (ii) a *goal-details* abstraction that constructs a tractable search space over undisclosed laundering transformation. (iii) **SDR**, a practical two-stage method that restores the effectiveness of standard detection methods even under laundering. Together, we hope this study establishes an actionable blueprint for data rights holders to verify unauthorized training under data laundering in black-box LLMs and *raises practitioners' awareness of data laundering*.

## 2 UNAUTHORIZED DATA DETECTION

Verifying the provenance of data used to train LLMs is a cornerstone of trustworthy AI, with critical implications for copyright compliance, data privacy, and license enforcement (Li et al., 2023a). The field has developed two main strategies for data governance, including proactive measures applied before/during training and post-hoc detection of trained models.

**Proactive defenses** are approaches that prevent or trace data misuse from the outset. *Data watermarking* embeds imperceptible signals, such as stylistic patterns, directly into training data, which can subsequently be detected in a model's outputs to establish provenance (Liang et al., 2024b). Differential privacy (Dwork, 2008), on the other hand, offers cryptographic-style guarantees against memorization by introducing calibrated noise during the training process, but frequently incurs a substantial penalty on model utility; it is rarely adopted in training LLMs where model performance is paramount. These are, however, voluntary disclosures and cannot verify the absence of undisclosed data sources, nor can they audit existing models whose provenance may be obscured. In brief, proactive defenses are essential but have inherent limitations; they are not implemented, and do not provide a mechanism for auditing existing models trained without such foresight.

**Post-hoc detection** seeks to determine whether specific data was used to train a deployed model *after* its development, often with only black-box access. This task is commonly instantiated via techniques derived from the membership-inference literature (Shokri et al., 2017), where the goal is

to distinguish training samples (members) from unseen samples (non-members) by exploiting statistical differences in model behavior (Li et al., 2025). Because overparameterized models tend to memorize their training data, they typically show higher confidence or lower loss on members (Carlini et al., 2021; 2022), a signal most post-hoc detectors rely on. Adapting these methods to modern LLMs is difficult due to the prohibitive cost of training shadow models (Carlini et al., 2022), so recent detectors instead analyze intrinsic signals from the target model itself, including loss-based (Ye et al., 2024), likelihood-based (Shi et al., 2023; Zhang et al., 2024), and calibrated-confidence approaches (Xie et al., 2024). Overall, these techniques aim to provide data rights holders with practical tools for detecting unauthorized data use and remain effective on standard benchmarks.

**Data laundering breaks post-hoc detection.** It is worth noting that all existing post-hoc detection methods are designed and evaluated under the "query with originals" regime, where the rights holder queries the *target* LLM with original proprietary texts and compares intrinsic-signal scores against a non-member corpus. This regime overlooks the pliability of natural language, creating a blind spot when the *target* LLM was trained on surrogates—semantics-preserving but stylistically or structurally altered variants generated via paraphrasing and back-translation (Barzilay & McKeown, 2001; Bannard & Callison-Burch, 2005), register/style transfer, or large-scale LLM-based rewriting (Witteveen & Andrews, 2019; Zeleke et al., 2025)—rather than on the originals. When exploited by model providers, this mechanism enables evasion of unauthorized-use detection, a practice we term *data laundering*. Since the *target* LLM never saw the originals, its memorization effect attaches to the surrogates, causing members to lose the intrinsic-score advantage on original texts. As a result, the score gap between members and non-members collapses, defeating standard detectors applied with originals.

**Can post-hoc detection be restored?** As aforementioned, the threat is compounded by the opacity of real-world LLM deployment. Data rights owners typically have only black-box access to the *target* LLM, and model providers rarely disclose training details. The specific laundering transformation, if one exists, is invisible to rights owners. Because the space of potential transformations is infinite, brute-force search is intractable, creating a practical impossibility for auditors: without knowing the hidden transformation, one cannot produce an *exact* training-time query that would elicit the memorization signal from the *target* LLM. Thus, rather than designing a new detector robust to laundered samples, we reframe the problem and focus on inferring the *laundering process* itself. By reverse-engineering the transformation's properties from the black-box behavior of the *target* LLM, we synthesize "training-like" queries that restore the statistical signals needed for detection. If successful, this approach re-enables standard off-the-shelf detectors, making them effective even under data laundering.

## 3    REVERSE-ENGINEERING THE LAUNDERING TRANSFORMATION

As established, directly finding the exact laundered data is intractable. Our approach, therefore, is to find a generative/synthesis process that *mimics* the unknown laundering transformation, leveraging a powerful auxiliary LLM as a controllable *transformation simulator* under prompt specification.

**Objective.** Given an off-the-shelf detector, a target LLM, an auxiliary LLM, and two corpora—the proprietary candidate set and the held-out non-training set—our goal is to identify a prompt-based synthesis transformation that maximizes the detector's score separation between the two sets. If the synthesized variants produced under a prompt yield a clear improvement in detection performance, this indicates that the target model was trained on laundered variants of the proprietary data, and that the prompt approximates the unknown laundering transformation used by the model provider. The detailed objective and threat model is shown in the Appendix. C

**Managing the prompt space.** Still, the search space of possible natural language prompts is effectively infinite (Zhang et al., 2025), making an unconstrained search infeasible. To solve this, we introduce a structured abstraction that makes the search tractable, leveraging established principles from prompt engineering (Mao et al., 2025) and linguistic theory (Agha, 2004; Myntti et al., 2025). We first structure the estimated transformation prompts using a *goal-details* schema, motivated by recent work showing that effective prompts can be decomposed into a core directive and supplementary modifiers such as context, constraints, or output format (Mao et al., 2025). We adapt this structure as follows: The *goal* defines the transformation's high-level intent and dominant stylistic shift (e.g., "rewrite in a lyrical style"); the *details* aggregate all other components that refine the out-

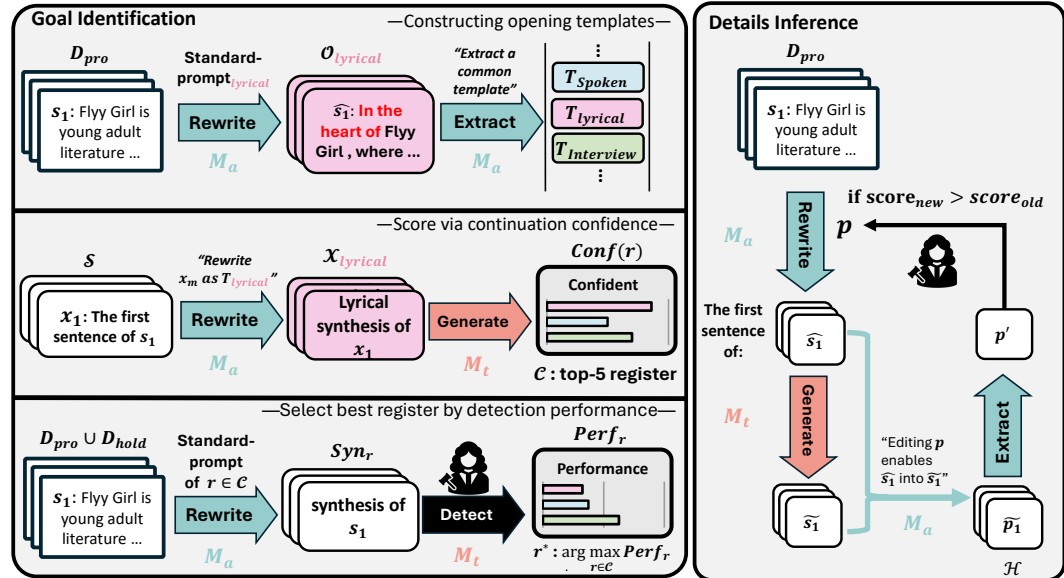

Figure 2: Pipeline of the **SDR** framework. In the **goal identification stage** (left part), **SDR** tries to find the register that is closely aligned with the laundering goal (See Algorithm. 1). The **details inference stage** (right part) tries to infer the remaining details in the laundering process (See Algorithm. 2).

put (e.g., "ensuring the imagery is vivid"). To further reduce the search space for *goals*, we ground it in an established linguistic taxonomy of 23 registers that collectively cover primary communicative forms (Henriksson et al., 2024). Our task now becomes selecting the most probable *goal* from this finite set of registers[3] and refining *details* within it. By combining the LLM-based transformation simulator with this structured *goal-details* prompt abstraction, we transform the intractable optimization problem into a constrained search. This leads to the two-stage method detailed next.

# 4    SYNTHESIS DATA REVERSION

We propose a two-stage framework, *synthesis data reversion* (**SDR**), to reverse the laundered data used to train the target model. The first stage, the *goal identification* stage, aims to infer the laundering *goal*. The second stage, *details inference*, aims to recover the supplementary conditions of the laundering process. Figure 2 overviews the pipeline, and a detailed description is in Appendix E.

**Goal identification stage.** At this stage, our method seeks to identify which of the 23 predefined registers best matches the laundering *goal* (Algorithm 1). A naïve approach would synthesize every proprietary sample into each register using the auxiliary LLM and evaluate which register yields the largest improvement in unauthorized data retention detection. However, synthesizing long-token sequences is costly. To reduce this cost, we rewrite only the opening sentence of each sample using register-specific opening templates. If the proprietary data were laundered toward a particular register, the laundered samples would likely adopt an opening resembling that register's template, leading the target model to generate continuations with higher confidence (Yeom et al., 2018). By measuring which register's opening sentences elicit higher continuation confidence, we can efficiently identify the register closest to the laundering goal.

Specifically, for **each register** $r$, we first use the auxiliary LLM to generate a *Standard-prompt* that can synthesize data into that register. Using Standard-prompt, the auxiliary LLM rewrites $n$ samples of $D_{\mathrm{pro}}$ and abstracts the first sentences of them into an opening template $\mathrm{T}_r$ (see constructing opening templates). Following the *Score via continuation confidence*, these templates are applied to rewrite the first sentence of the original data (i.e., the first sentence of $s \in \mathrm{UniformSample}(D_{\mathrm{pro}}, m)$). Rewritten sentences are then provided to $M_t$ and measure the average model's continuation confidence $\mathrm{Conf}(r)$ (Details of the confidence calculation are shown in Appendix B). The top-5 registers with the highest $\mathrm{Conf}(r)$ are retained as possible registers $C$.

---

[3]We acknowledge that this taxonomy was not designed for data laundering and thus has limitations, which we discuss in Appendix K.

---

**Algorithm 1** Goal identification stage

---

**Require:** Proprietary originals $D_{\text{pro}}$, held-out data $D_{\text{held}}$, target model $M_t$, auxiliary LLM $M_a$, set of 23 registers $R$, sample size $n$ and $m$

**Ensure:** the register $r^* \in R$ that is closely aligned with the laundering directive.

1: **—Constructing opening templates—**
2: **for all** $r \in R$ **do**
3:     Standard-prompt$_r \leftarrow M_a$("Give me a prompt that can transfer text into register $r$")
4:     $O_r \leftarrow \{$ The first sentence of $M_a$(Standard-Prompt$_r, s) \mid s \in \text{UniformSample}(D_{\text{pro}}, n) \}$
5:     $\text{T}_r \leftarrow M_a$("Extract a common template.", $O_r$)
6: **end for**
7: **—Score via continuation confidence—**
8: **for all** $r \in \mathcal{R}$ **do**
9:     $S \leftarrow \{$ The first sentence of $s) \mid s \in \text{UniformSample}(D_{\text{pro}}, m) \}$
10:     $X_r \leftarrow \{ M_a$("Rewrite $x$ as $\text{T}_r$") $\mid x \in S \}$
11:     **for** $j \leftarrow 1$ **to** $m$ **do**
12:         $c_j \leftarrow$ Average next token confidence of $M_t(X_r[j])$
13:     **end for**
14:     $\text{Conf}(r) \leftarrow \frac{1}{m} \sum_j c_j$
15: **end for**
16: $C \leftarrow$ top-5 registers with largest $\text{Conf}(r)$
17: **—Select best register by detection performance—**
18: **for all** $r \in C$ **do**
19:     $\text{Syn}_r \leftarrow \{ M_a(\text{Standard-prompt}_r, d) \mid d \in D_{\text{pro}} \cup D_{\text{held}} \}$
20:     $\text{Perf}_r \leftarrow$ Unauthorized training data detection on $M_t$ using $\text{Syn}_r$
21: **end for**
22: $r^* \leftarrow \arg\max_{r \in C} \text{perf}_r$
23: **return** $r^*$, Standard-prompt$_{r^*}$

---

Finally, the closest register $r^*$ is selected from $C$ based on its unauthorized training data detection performance (see "Select best register by detection performance").

---

**Algorithm 2** Details inference stage

---

**Require:** register $r^*$ and Standard-prompt$_{r^*}$ got from Algorithm 1 , proprietary data $D_{pro}$, held-out data $D_{\text{held}}$, target model $M_t$, auxiliary LLM $M_a$, iteration budget $K$, sample size $l$

**Ensure:** Reversed prompt

1: $p = $ Standard-prompt$_{r^*}$
2: **function** CONDITIONINFERENCE($D_{\text{pro}}, p, M_t, M_a$)
3:     **for all** $s \in \text{UnifiedSample}(D_{\text{pro}}, l)$ **do**
4:         $\hat{s} \leftarrow M_a(p, s)$, $\tilde{s} \leftarrow M_t$(the first sentence of $\hat{s}$)
5:         $H.\text{APPEND}(M_a$("Editing $p$ enables the transofrmation of $\hat{s}$ into $\tilde{s}$"))
6:     **end for**
7:     **return** $M_a$("Extract a common prompt.", $H$)
8: **end function**
9: **function** EVALUATE($D_{pro}, D_{\text{held}}, p, M_t, M_a$)
10:     $\text{Syn}_p \leftarrow \{ M_a(p, x) \mid x \in D_{\text{pro}} \cup \mathcal{D}_{\text{held}} \}$
11:     $\text{Perf}_r \leftarrow$ Unauthorized training data detection on $M_t$ using $\text{Syn}_p$
12:     **return** $\text{Perf}_p$
13: **end function**
14: **for** $k \leftarrow 0$ **to** $K - 1$ **do**
15:     $p' \leftarrow$ CONDITIONINFERENCE($D_{\text{pro}}, p, M_t, M_a$)
16:     $\text{score}_{\text{new}} \leftarrow$ EVALUATE($D_{\text{pro}}, D_{\text{held}}, p', M_t, M_a$), $\text{score}_{\text{old}} \leftarrow$ EVALUATE($D_{\text{pro}}, D_{\text{held}}, p, M_t, M_a$)
17:     **if** $\text{score}_{\text{new}} > \text{score}_{\text{old}}$ **then**
18:         $p \leftarrow p'$
19:     **end if**
20: **end for**
21: **return** $p$

---

**Details inference stage.** Once the closest register has been identified, we can reverse the laundered data by synthesizing the original samples into that register. However, this reversion may still diverge from the true laundered training data, as additional *details* may have been applied in the laundering process. The second stage seeks to recover these additional details (see Algorithm 2). Directly comparing the closest register synthesis with the true training data would reveal such details, but

Table 2: The average performance of each unauthorized training data detection method across data synthesized from **different inside and outside luandering process**. The experience is located on Pythia-6.9B (Biderman et al., 2023), fine-tuned on Wikipedia synthesis. The results for each prompt are provided in Appendix J. We report the additional TPR@1% results in Appendix R.

| Method | Inside registers | | | Outside registers | | |
|---|---|---|---|---|---|---|
| | AUC | ACC | TPR@5% | AUC | ACC | TPR@5% |
| Recall. | 64.7% | 63.4% | 8.9% | 61.7% | 61.4% | 5.6% |
| Recall+**SDR** | **76.2%** | **72.0%** | **25.3%** | **73.4%** | **73.3%** | **23.2%** |
| Loss. | 63.7% | 62.8% | 10.7% | 62.7% | 62.6% | 9.2% |
| Loss+**SDR** | **76.6%** | **72.6%** | **26.2%** | **75.5%** | **75.5%** | **22.9%** |
| Ref | 68.6% | 67.0% | 15.0% | 67.6% | 65.6% | 13.2% |
| Ref+**SDR** | **74.8%** | **70.8%** | **29.9%** | **72.0%** | **72.1%** | **24.2%** |
| Zlib | 63.9% | 63.5% | 15.2% | 63.6% | 63.5% | 13.9% |
| Zlib+**SDR** | **68.9%** | **66.7%** | **18.8%** | **68.4%** | **68.4%** | **16.2%** |
| Min-K | 63.5% | 62.6% | 11.8% | 64.2% | 62.5% | 10.5% |
| Min-K+**SDR** | **75.1%** | **71.6%** | **25.1%** | **73.6%** | **73.5%** | **22.7%** |

this is infeasible for the data rights owner unless similar data can be found. In the previous stage, we know that the first sentence of the closest register synthesis resembles that of the target model's training data. Providing such a familiar opening sentence to the target model activates its memory of training data, enabling it to reproduce the corresponding memorized continuations that are similar to the training data. As a result, analyzing the differences between the closest register synthesis and the reproduced continuations enables us to recover the additional conditions.

Particularly, we first synthesize proprietary samples with the auxiliary LLM using an initial prompt $p$ that can synthesize data into the closest register (the Standard-prompt$_{r*}$ got from the previous stage). Using the function CONDITIONINFERENCE, the first sentence of the synthesis is fed into the target model to generate continuations. Both the synthesized data and the generated continuations are provided to $M_a$, which infers the details involved in the laundering process. We apply CONDITIONINFERENCE to multiple samples from $D_{pro}$, obtaining a collection of candidate prompts $H$. We then query $M_a$ with these prompts to distill a common system prompt that captures their shared transformation pattern. To determine whether refinement improves performance, we apply the EVALUATE function; if so, the refined prompt replaces the initial one and the process continues iteratively. Through iterative updates, a refined prompt with enhanced details is created that best approximates the laundering process, improving detection performance on its reversed data.

## 5 EXPERIMENTS AND RESULTS

**Dataset and victim models.** We utilize the MIMIR (Deng et al., 2023) benchmark dataset (a detailed introduction to the MIMIR dataset is provided in Appendix F.), a widely recognized resource in research on unauthorized training data detection. To evaluate the generality of our method, we select three subsets from MIMIR: Wikipedia, C4, and HackerNews, corresponding to encyclopedia articles, web text, and news reports, respectively. As victim models, we employ several different architectures, including Pythia (Biderman et al., 2023), Falcon (Zhang et al., 2022), and LLaMA-2 (Touvron et al., 2023) to evaluate the robustness of **SDR** across architecture.

**Baselines and metrics.** We involve five baseline unauthorized training data detections in our experiments: Loss (Yeom et al., 2018), which uses likelihood loss as the membership score; Ref (Carlini et al., 2022), which calibrates input loss via a reference model; Zlib (Carlini et al., 2021), which compresses input loss through entropy coding; Min-K% (Shi et al., 2023) and RecaLL (Xie et al., 2024) as introduced in Section 2. Following prior work in (Carlini et al., 2022), we report three metrics: Area Under the Curve (AUC), Attack success rate (ASR), and True Positive Rate at 5% False Positive Rate (TPR@5%). Details explaining the metrics are shown in Appendix G.

**Evaluation settings and implementation details.** We evaluate the effectiveness of **SDR** by examining whether it reverses data to enhance the performance of existing unauthorized training data detection methods against target LLMs trained on laundered data. (See Appendix H for details.)

**Synthesized prompt setting.** We consider two types of prompts that may be applied by the model provider: inside-register and outside-register. Inside-register prompts assume that the model provider synthesizes the original data into one of the 23 sub-registers. For each register, we use GPT-5 (Leon, 2025) to generate a corresponding prompt. Outside-register prompts are those generated

Table 3: Comparison of average performance of unauthorized training data detection across **different datasets** trained with synthesis using outside register prompts.

| Method | Wikipedia | | | Hackernews | | | C4 | | |
|---|---|---|---|---|---|---|---|---|---|
| | AUC | ACC | TPR@5% | AUC | ACC | TPR@5% | AUC | ACC | TPR@5% |
| Recall | 61.7% | 61.4% | 5.6% | 53.9% | 56.2% | 9.9% | 52.2% | 55.0% | 5.0% |
| Recall+**SDR** | **73.4%** | **73.3%** | **23.2%** | **61.6%** | **60.3%** | **10.7%** | **65.2%** | **63.0%** | **12.0%** |
| Loss | 62.7% | 62.6% | 9.2% | 54.2% | 56.0% | 6.3% | 58.4% | 59.1% | 7.2% |
| Loss+**SDR** | **75.5%** | **75.5%** | **22.9%** | **62.7%** | **59.6%** | **8.7%** | **67.3%** | **64.2%** | **13.0%** |
| Min-K | 64.2% | 62.5% | 10.5% | 53.8% | 55.9% | 5.7% | 57.6% | 59.2% | 6.2% |
| Min-K+**SDR** | **73.6%** | **73.5%** | **22.7%** | **61.7%** | **60.8%** | **8.3%** | **66.8%** | **64.7%** | **11.8%** |

Table 4: Comparison of average performance of unauthorized training data detection across three **different model architectures** fine-tuned with synthesis generated by outside register prompts.

| Method | Pythia-6.9B | | | Falcon-7B | | | LLaMA-2-7B | | |
|---|---|---|---|---|---|---|---|---|---|
| | AUC | ACC | TPR@5% | AUC | ACC | TPR@5% | AUC | ACC | TPR@5% |
| Recall | 61.7% | 61.4% | 5.6% | 62.1% | 61.8% | 8.2% | 61.6% | 61.6% | 8.2% |
| Recall+**SDR** | **73.4%** | **73.3%** | **23.2%** | **72.4%** | **69.1%** | **23.0%** | **67.5%** | **66.2%** | **12.5%** |
| Loss | 62.7% | 62.6% | 9.2% | 64.4% | 64.4% | 11.5% | 64.5% | 64.4% | 11.1% |
| Loss+**SDR** | **75.5%** | **75.5%** | **22.9%** | **71.2%** | **68.0%** | **20.2%** | **73.6%** | **70.4%** | **26.9%** |
| Min-K | 64.2% | 62.5% | 10.5% | 63.3% | 62.6% | 10.5% | 62.5% | 62.5% | 13.9% |
| Min-K+**SDR** | **73.6%** | **73.5%** | **22.7%** | **70.2%** | **68.0%** | **20.0%** | **70.9%** | **68.2%** | **22.9%** |

by GPT-5 that do not align with any established register. The full list of inside- and outside-register prompts is provided in Appendix I.

**Result analysis across different synthesized prompts.** To evaluate whether **SDR** can reverse synthesize training data from different prompts, we use GPT-4o (Roumeliotis & Tselikas, 2023) to synthesize the MIMIR-Wikipedia data into new data with different inside- and outside prompts (mentioned in Section 5) and fine-tune a Pythia-6.9B (Biderman et al., 2023) model. Table 2 shows that across both inside- and outside-register prompts, **SDR** consistently enhances the average performance of the detection. Specifically, the average detection AUC of Loss increases by 12.9% across the inside prompts and 12.8% across the outside prompts. The specific results for each prompt are provided in Appendix J. In addition, we have conducted experiments on mixed-register transformations, where two registers are combined within the laundering prompt; the corresponding results are provided in Appendix O.

**Result analysis across different datasets.** To evaluate the robustness of **SDR** across different datasets, we applied unauthorized training data detections to Pythia-6.9B models, which were trained with synthesized data from various datasets (Wikipedia, Hackernews, and C4) under outside prompts. Table 3 shows that across all three datasets, **SDR** consistently improves detection performance. For example, Recall achieves a clear AUC gain on all datasets corresponding to an average improvement of 10.8%.

**Result analysis across different LLM structures.** To evaluate the robustness of **SDR** across different trained model architectures, Table 4 shows the performance of unauthorized training data detections on three different model architectures (Pythia-6.9B, Falcon-7B, and LLaMA-2-7B) fine-tuned with MIMIR-Wikipedia synthesis using outside register prompts. Across all three models, **SDR** consistently enhances detection effectiveness. For example, Recall achieves substantial AUC gains on all models, with an average improvement of 9.3%.

**Result analysis with different auxiliary models.** We examine a scenario in which the auxiliary LLM applied by the data rights holder for **SDR** differs from the one employed by the model provider used to launder data. As shown in Table 5, in this experiment, we consider that the model provider synthesizes data using GPT-4o, while **SDR** employs auxiliary LLMs such as Claude and DeepSeek for reverse synthesis. Results are averaged using the first ten inside prompts. From the result, we can find that **SDR** achieves an average AUC improvement of 13.5% on GPT-4o, 11.5% on Claude, and 12.7% on DeepSeek. These increasing values are close, indicating that **SDR**'s effectiveness is stable across different auxiliary LLMs. We also evaluate cross-launderer transferability using two forms of third-party laundering pipelines—specifically, data laundered by DeepSeek-v3 and a human-rewritten corpus. Detailed experimental settings and results are provided in Appendix N.

Table 5: Comparison of the average performance of unauthorized training data detection with **SDR** using **different auxiliary LLMs**.

| | GPT-4o | | | Claude | | | DeepSeek | | |
|---|---|---|---|---|---|---|---|---|---|
| Method | AUC | ACC | TPR@5% | AUC | ACC | TPR@5% | AUC | ACC | TPR@5% |
| Recall | 64.5% | 63.5% | 9.9% | 66.8% | 65.9% | 9.6% | 65.5% | 64.8% | 7.4% |
| Recall+**SDR** | **79.7%** | **75.0%** | **31.6%** | **79.8%** | **75.5%** | **31.2%** | **79.8%** | **75.5%** | **31.2%** |
| Loss | 63.4% | 62.2% | 14.3% | 67.3% | 65.3% | 12.1% | 65.9% | 64.7% | 12.2% |
| Loss+**SDR** | **80.3%** | **75.6%** | **32.6%** | **79.1%** | **74.7%** | **32.8%** | **79.0%** | **74.7%** | **32.8%** |
| Min-K | 63.7% | 62.3% | 11.0% | 68.0% | 65.5% | 14.5% | 67.0% | 64.6% | 14.2% |
| Min-K+**SDR** | **72.0%** | **75.1%** | **32.8%** | **77.8%** | **73.1%** | **30.3%** | **77.8%** | **73.1%** | **30.3%** |

**Ablation study.** To assess the contribution of each stage in the proposed **SDR** framework, we conduct an ablation study by selectively removing individual stages. By comparing the performance of detections on the data reversed with the full **SDR** framework against that with individual stages, we can evaluate the necessity of each stage. As shown in Figure 3, directly applying the identified directive from the *goal identification stage* (w/o stage 2) to reverse the synthesized data leads to degradations in all average detection metrics across different inside prompts. In particular, with TPR@5% dropping by 7.5% compared to the full **SDR** framework (SDR). Skipping *the register identification stage* and only relying on the *details inference* (w/o stage 1) causes even more severe degradation, reducing TPR@5% by 12.5%. These results demonstrate that both stages are indispensable. We also perform an ablation study to analyze the behavior of **SDR** under different configurations of the hyperparameters $K$, $l$, $m$, and $n$, as reported in Appendix P.

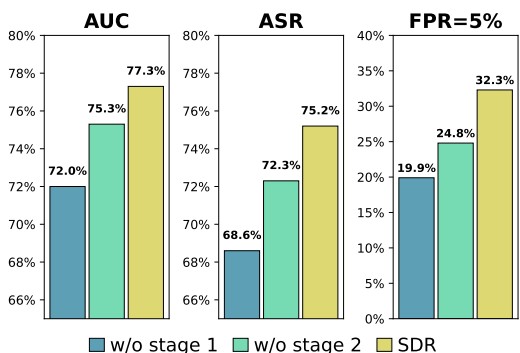

Figure 3: Ablation study on the effectiveness of each stage in **SDR**. Results are reported as the average performance of unauthorized training data detections across different inside prompts. Removing the directive identification stage (w/o stage 1) or the detailed prompt condition inference stage (w/o stage 2) leads to noticeable degradation, while the full **SDR** consistently achieves the best performance.

**Negative-control experiment.** To address the concern of whether **SDR** triggers a false alarm (i.e., improve $\text{Pref}_p$ even when the model provider did not launder $D_{pro}$.), we add a negative-control experiment where neither $D_{pro}$ nor its laundered variants were used. Experimental results and implementation details are provided in Appendix M.

**Comparison to reverse prompt engineering methods.** To contextualize **SDR** within the broader landscape of prompt-search and reverse-engineering approaches, we provide in Appendix Q a comparison between **SDR** and state-of-the-art reverse prompt engineering methods (Li & Klabjan, 2024), including an adapted version of that aligns with our problem setting.

**Partial laundering experiment.** In real-world scenarios, the model provider may launder only a subset of $D_{pro}$ for training. To demonstrate that **SDR** does not rely on full laundering, we additionally conduct experiments showing that SDR remains effective under partial laundering. The results are reported in Appendix S.

## 6 CONCLUSION

This paper identified a critical vulnerability in current auditing practices: conventional unauthorized training data detections fail under data laundering, leaving a loophole that enables model providers to obscure the provenance of training data. To address this challenge, we proposed **SDR**, a two-stage framework that reconstructs the synthesis process by inferring a prompt to recover laundered data, thereby restoring the detectability of unauthorized usage. Through extensive evaluation across datasets, model architectures, and auxiliary LLM models, we demonstrated that **SDR** enhances the effectiveness of unauthorized training data detection. In future work, the focus should be on developing finer-grained, task-specific directive taxonomies to improve the accuracy and robustness of prompt reversal. In sum, we believe **SDR** opens a promising direction for developing robust privacy auditing tools against data laundering.

# 7 ETHICS STATEMENT

This work focuses on defending against unauthorized data laundering in LLM training by restoring the effectiveness of post-hoc detection. All experiments were conducted on publicly available datasets, and no proprietary or personal data was used. While there is a risk that our techniques could be misused to improve laundering attacks, our contributions are explicitly framed for defensive purposes, aiming to strengthen accountability and responsible governance in AI systems.

# 8 REPRODUCIBILITY STATEMENT

We have made significant efforts to ensure the reproducibility of our results. All datasets used in this paper are publicly available, and details of data synthesis procedures are provided in Appendix I. Complete descriptions of model architectures, training settings, and evaluation protocols are included in the main paper and Appendix H. For each experiment, we specify hyperparameters, implementation details, and the auxiliary LLM prompts used for synthesis and reverse synthesis. Our codebase, built on PyTorch and Hugging Face Transformers, was included in the submitted supplementary material and will be released upon publication to facilitate full replication of our experiments.

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

## A    REGISTER TAXONOMY

The register taxonomy proposed by Henriksson et al. (2024) defines 23 sub-registers across nine major categories, ranging from narrative forms (e.g., news and sports reports, blogs) and informational texts (e.g., encyclopedia entries, research articles, legal documents) to opinion pieces, persuasive descriptions, interactive discussions (e.g., FAQs, interviews), instructional texts (e.g., recipes), spoken and lyrical registers, and machine-translated content (details are shown in Table 6). This taxonomy provides near-comprehensive coverage of open-domain texts, offering a systematic and scalable framework that reduces unbounded stylistic variation into a bounded set of functional categories.

## B    AVERAGE TOKEN-LEVEL CONFIDENCE

We use the average token-level confidence to evaluate the model's confidence. We define the confidence score of register $r$ as

$$\text{Conf}(r) = \frac{1}{n} \sum_{j=1}^{n} \left( \frac{1}{L_j} \sum_{i=1}^{L_j} \max_{w \in V} P_M\left( w \mid X_{r,<i}^{(j)} \right) \right),$$

where $r$ is the register, $V$ is the vocabulary, $L_j$ is the length of the generated continuation for the rewritten opening sentence, and $X_{r,<i}^{(j)}$ refers to the concatenation of the rewritten opening sentence and the first $i-1$ tokens already generated in the continuation, $M$ represents the target model.

Table 6: Register categories and abbreviations

| Name | Abbr. | Name | Abbr. |
|------|-------|------|-------|
| **Lyrical** | LY | Encyclopedia article | en |
| **Spoken** | SP | Research article | ra |
| Interview | it | Description of a thing or person | dtp |
| **Interactive discussion** | ID | FAQ | fi |
| **Narrative** | NA | Legal terms & conditions | lt |
| News report | ne | **Opinion** | OP |
| Sports report | sr | Review | rv |
| Narrative blog | nb | Opinion blog | ob |
| **How-to or instructions** | HI | Denominational religious blog or sermon | rs |
| Recipe | re | Description with intent to sell | ds |
| **Informational persuasion** | IP | Informational description | IN |
| News & opinion blog or editorial | ed | | |

## C  THE OBJECTIVE AND THREAT MODE

**Threat model.**  Our setting follows a black-box auditing formulation used in post-hoc unauthorized data detection (Xie et al., 2024; Carlini et al., 2022), describing a threat model where: 1. The model provider trains the target LLM $M_t$ on an unknown training set $D_{\text{train}}$ that may include laundered variants of some samples in a proprietary dataset $D_{\text{pro}}$. 2. The data rights holder owns a proprietary dataset $D_{\text{pro}}$ and has a held-out reference set $D_{\text{held}}$ guaranteed not to appear in $D_{\text{train}}$. The rights holder only has access to (i) $D_{\text{pro}}$, (ii) $D_{\text{held}}$, and (iii) black-box query access to $M_t$ (e.g., API). It does **not** have access to $D_{\text{train}}$, nor to the (potentially unknown) laundering pipeline. 3. An auxiliary LLM $M_a$ is any model that the rights holder can query to synthesize surrogate samples. We additionally summarize all symbols and their roles in Table 7.

| Symbol | Description & Role | Visible to Rights Holder? |
|--------|-------------------|---------------------------|
| $M_t$ | Target LLM trained on $D_{\text{train}}$ | No |
| $M_a$ | Auxiliary LLM used only for synthesis | Yes |
| $T$ | True (unknown) laundering transformation | No |
| $p$ | Reverse-synthesis prompt inferred by **SDR** | Yes |
| $D_{\text{pro}}$ | Proprietary dataset (candidate corpus) | Yes |
| $D'_{\text{pro}}$ | Unknown in-training subset of $D_{\text{pro}}$ | No |
| $D_{\text{train}}$ | Model provider's full training dataset | No |
| $D_{\text{lau}}$ | Laundered version of $D'_{\text{pro}}$ | No |
| $D_{\text{held}}$ | Held-out non-training reference dataset | Yes |
| $s$ | Original sample | Yes |
| $\hat{s} = M_a(p, s)$ | Synthetic surrogate rewrite | Yes |
| $\text{Syn}_p(S)$ | Surrogate set induced by prompt $p$ | Yes |
| $\text{Perf}_p$ | Detector performance under prompt $p$ | Yes |

Table 7: Notation table for the threat model.

**What SDR optimizes.**  **SDR** seeks to optimize a prompt $p$ so that samples in $D_{\text{pro}}$ are mapped closer to the true laundered variants used in training $M_t$, thereby restoring the effectiveness of standard unauthorized training-data detectors. Firstly, Uses $M_a$ to synthesize $\text{Syn}_p(D_{pro})$ and $\text{Syn}_p(D_{held})$. And then Runs an off-the-shelf detector (ReCaLL, Min-K, etc.) on $M_t$ using these two sets, exactly as in standard post-hoc unauthorized data detection but using the synthesized variants. Finally, **SDR** takes the detector's output scalar score (e.g., ReCaLL's two-sample score) as $\text{Perf}_p$, and **SDR** seeks to optimize a prompt $p$ that maximize $\text{Perf}_p$.

**Detail introduction about $\text{Perf}_p$.**  Given a fixed detector (e.g., ReCaLL, Min-K), a prompt $p$ induces synthetic surrogates $\text{Syn}_p(D_{\text{pro}})$ and $\text{Syn}_p(D_{\text{held}})$ via $M_a$. We then run the detector on $M_t$

precisely as in prior work, but using these surrogates as candidate vs. reference sets, and record a scalar performance measure $\text{Pref}_p$. Algorithms 2 and 1 describe our pipeline that searches over the prompt space to maximize this scalar. $\text{Pref}_p$ distinguishes $\text{Syn}_p(D_{\text{pro}})$ from $\text{Syn}_p(D_{\text{held}})$, implying that $M_t$ was indeed trained on a laundered version of $D_{\text{pro}}$, and $p$ can recover the unknown laundering process used by the model provider. Otherwise, it implies that the current prompt $p$ cannot recover the laundered training data, and we need to continue searching for more plausible ones. Eventually, if no such performance-improving prompt can be found, we consider that no laundering of $D_{\text{pro}}$ was used in training $M_t$.

## D  OPENING TEMPLATE

Table 8 presents the representative opening templates $T_r$ that we derived for each register. These templates were obtained by synthesizing a small subset of samples into the corresponding register $r$ and then prompting a large language model to extract a generalized first-sentence structure. As shown in the table, each register exhibits distinct stylistic cues in its openings: for example, lyrical texts often begin with abstract imagery, interviews with a direct address from the interviewer, and storytelling narratives with a scene-setting phrase such as "Once upon a time." Such templates capture the prototypical entry points of different registers, which serve as useful anchors for aligning synthesized outputs with their intended discourse forms.

## E  MATHEMATICAL NOTATION

We first summarize the key mathematical symbols used throughout the paper.

| Symbol | Description |
|---|---|
| $\mathcal{R}$ | Set of 23 establised registers. |
| $r^*$ | Register selected as most closely aligned with the laundering goal. |
| $n, m, l$ | Sample sizes used in constructing templates, scoring, and inference, respectively. |
| $K$ | Maximum number of iterations in the detailed prompt condition inference stage. |
| $\text{Standard-prompt}_r$ | A canonical prompt that synthesizes text into register $r$. |
| $T_r$ | Opening template extracted for register $r$. |
| $\hat{s}$ | Synthetic data generated by $M_a$ from an original sample $s$ under prompt $p$. |
| $\tilde{s}$ | Continuation produced by the target model $M_t$ when prompted with $\hat{s}$. |
| $\text{Conf}(r)$ | Average next-token confidence of $M_t$ under register $r$. |
| $\mathcal{C}$ | Candidate set of top-$k$ registers with highest confidence scores. |
| $\text{Syn}_r$ | Dataset synthesized into register $r$ by $M_a$. |
| $\text{Perf}_r$ | Unauthorized training data detection performance of $\text{Syn}_r$ on $M_t$. |
| $p$ | Current reverse-synthesis prompt refined during iterations. |

## F  A INTRODUCTION TO THE MIMIR DATASET

The MIMIR benchmark is specifically constructed for evaluating unauthorized training data detection. For each model to be evaluated (Pythia, Falcon, LLaMA-2), it defines "seen" splits (i.e., ground-truth training data) against "non-seen" splits (i.e., non-training data), which contain documents collected after the model's release and thus not used in its pretraining (Deng et al., 2023). This therefore ensures the "non-seen" splits thus can be reliably used in our evaluation. The MIMIR dataset is publicly available at https://huggingface.co/datasets/iamgroot42/mimir.

## G  EVALUATION METRICS

Following Carlini et al. (Carlini et al., 2022), we adopt three complementary metrics to evaluate membership inference attacks.

Table 8: Registers and their corresponding opening templates

| Register | Template (first-sentence / opening) |
|---|---|
| Lyrical | In the heart of [abstract domain], a tale unfolds, where [abstract concept], [abstract detail], [abstract entity], [abstract action]. |
| Spoken style | So, let's talk about [TOPIC]. |
| Interview | **Interviewer:** Thank you for joining us, [Person/Expert Title]. Can you tell us about [Subject/Topic]? |
| Interactive discussion | **[Participant 1]:** So, have you guys heard about [Topic/Subject]? I recently came across some interesting information about it. |
| Storytelling narrative | Once upon a time, in a [adjective] [type of place] called [place name], there lived a [adjective] [type of character] named [character name]. |
| News report | **[Event/Topic]: [Description/Significance] [Location/Context]** – [Details about the subject, including noteworthy contributions, roles, or milestones]. |
| Sports report | In a thrilling [event/display/action], [subject/actor] has [verb] [description/impact] in [field/area/genre]. |
| Narrative blog post | In the context of [broad category or field], [subject or specific work] has made a significant impact, often leading to [general observation or effect]. |
| Step-by-step guide (How-to) | **Step-by-Step Guide to Understanding [Subject]** — Step 1: [Initial focus or background]. Learn that [Subject Description]. |
| Recipe | **Recipe for [General Concept]: [Specific Edition/Style]** — *Ingredients:* [Variable 1], [Variable 2], [Variable 3]. . . |
| Encyclopedia article | **[Subject]** is a [type/category] that [provides a description or function], [additional information if applicable]. |
| Research article | This article explores the significance of [subject or topic], a [description or classification], characterized by [notable features or contributions]. |
| Description of a thing or person | Introducing [Subject/Entity], a [descriptor] [type/category] [context/detail] renowned for its [property/characteristic]. |
| FAQ | **What is [Subject]?** — [Subject] is a [general category or description] [specific type or detail] [additional information]. |
| Legal terms & conditions | **Terms and Conditions Regarding [Subject/Theme]**. |
| Opinion | In my view, [Subject/Entity] represents [significance/impact/legacy] in [field/area], and its influence on [audience/community/context] cannot be overstated. |
| Review | [Subject] is a [descriptor] that [verb phrase] [contextual information]. |
| Opinion blog (editorial) | When we think of [general category or field], [a notable example or subject] often comes to mind. |
| Denominational religious sermon | Beloved congregation, today we gather to reflect upon [individual/concept] that illuminates our lives and encourages us to contemplate our shared journey. |
| Description with intent to sell | Introducing [Subject]: a [descriptor] [product/service] designed for [use case]; discover how it [benefit/outcome] for [target user]. |
| Informational persuasion | In the context of [domain or field], few [types/categories] resonate as profoundly within [subfields] as [specific work/name/entity]. |
| Informational description | [Entity/Subject] is a [description] in the field of [broader category], specifically within [subcategory/locale]. |
| News & opinion blog or editorial | When we think of [general category or field], [notable subject] often comes to mind — situating today's discussion of [topic] within [context]. |

**Area Under the ROC Curve (AUC).** AUC measures the overall discriminative power of the attack, independent of any specific threshold. It reflects how well an unauthorized training data detection method can separate training data from unseen data on average. It may overstate effectiveness since it also includes high false-positive regions that are less relevant in practice.

**Attack Success Rate (ASR).** ASR measures the fraction of correctly identified training data under a single decision threshold that maximizes balanced accuracy across training data and unseen data. Unlike AUC, ASR reflects the practical effectiveness of an attack when deployed, as real-world unauthorized training data detentions typically operate at a single fixed threshold.

**True Positive Rate at 5% False Positive Rate (TPR@5%).** This metric evaluates the ability of a detection to identify training data while maintaining a strict false-positive constraint. Prior work highlights that low false-positive regimes are the most meaningful for privacy evaluation, since even a small number of incorrect training data decisions can undermine the credibility of the attack. TPR@5% therefore provides a high-precision view of attack success.

## H    DETAILS OF IMPLEMENTATION AND EVALUATION

We evaluate the effectiveness of our approach by examining whether the data reversed by **SDR** enhances the performance of existing unauthorized training data detection methods against LLMs trained on synthesized data. Specifically, we first sample 200 data points from the dataset that have not been seen by the *target* LLM. These 200 samples are synthesized into a new version using a prompt to simulate the data laundering process applied by the model provider. We then randomly select half of these synthesized samples as training data to fine-tune the LLM (training details are provided in Table 9), while the remaining half serves as non-training data. Subsequently, **SDR** is employed to infer a transformation prompt that restores the synthesis process and recovers the synthesized samples. Using this inferred prompt, we rewrite the original 200 data. Finally, we apply baseline unauthorized training data detections to both the original 200 samples and the 200 samples rewritten with the inferred prompt, comparing their attack performance to assess whether the inferred prompt improves the effectiveness of detections.

| Lora_alpha | r | epochs | lr | gradient accumulation steps | weight_decay | logging_strategy |
|---|---|---|---|---|---|---|
| 32 | 8 | 60 | 0.0004 | 25 | 0.1 | "epoch" |

Table 9: Hyperparameter settings for LoRA fine-tuning.

## I    SYNTHESIZED PROMPTS USED IN EXPERIENCES

Table 10 lists the complete set of inside-register prompts corresponding to the 23 predefined registers in our framework. Each prompt is generated by GPT-5 using the instruction: "Generate a rewriting prompt that transfers the text into [REGISTER]." Here, [REGISTER] denotes one of the 23 registers. Table 11 lists the complete set of outside-register prompts; none align with any of the 23 predefined registers. These prompts are generated by GPT-5 using the instruction: "You are a prompt generator. Generate rewriting prompts that transform the original content into broadly different domains (e.g., Facebook post, academic article, children's story, legal document). Make the prompts as mutually distinct in domain as possible, and ensure each induces substantial changes to the original text rather than merely surface-level edits."

## J    THE SPECIFIC RESULTS FOR EACH PROMPT

Table 2 reports the average performance of each unauthorized training data detection method across models fine-tuned on data synthesized with different prompts. Tables 12 and 13 present the detailed average performance across five unauthorized training data detection methods for each inside-synthesized prompt. Tables 14 present the details for each outside-synthesized prompt. The "Original Prompt" column denotes the true synthesized prompt. The "Reversed Prompt" column denotes the best prompt recovered by **SDR**. The "Orig." column reports the average AUC across detection methods using the original data, whereas the "**SDR**" column reports the results using **SDR**-reversed data.

### J.1    ANALYSIS OF NON-GOAL SYNTHESIS PROCESS

We analyze a special case where the synthesis prompt does not provide an explicit directive. For example, consider the outside-synthesized prompt: "Rewrite the text with stronger transitions between sentences and paragraphs, ensuring smoother reading without adding new information." Al-

Table 10: Inside register prompts

| ID | Synthesize Prompts |
|---|---|
| 1 | Rewrite the text in a lyrical style, ensuring the imagery is vivid, the rhythm flows naturally. |
| 2 | Rewrite the text in a spoken style, making it sound natural and conversational, and ensure the tone feels engaging and easy to follow for a live audience. |
| 3 | Rewrite the text in the form of an interview, ensuring the questions flow naturally and the answers provide clear, engaging explanations for the audience. |
| 4 | Rewrite the text as an interactive discussion between two or more participants, ensuring the conversation flows logically, with each speaker's tone and style clearly distinguishable. |
| 5 | Rewrite the text as a storytelling narrative. The story should flow naturally, use simple and engaging language, and be easy for all kinds of listeners to follow. |
| 6 | Rewrite the text in the style of a news report, ensuring the information is presented objectively and concisely. |
| 7 | Rewrite the text as a sports report, ensuring the action is described with dynamic, energetic language that conveys the pace, tension, and excitement of the event. |
| 8 | Rewrite the text as a narrative blog post, organized into clear sections with subheadings. Use a tone that is engaging and reflective, blending storytelling with explanation. |
| 9 | Rewrite the text as a step-by-step instructional guide. Break the content into numbered steps, with each step beginning with a clear imperative verb. |
| 10 | Rewrite the text as a recipe, introduce the information as sequential steps. |
| 11 | Rewrite the text to persuade the reader through factual information, making sure to include at least three specific data points or statistics to support the argument. |
| 12 | Rewrite the text as a sales description, and be sure to include a clear call-to-action at the end. |
| 13 | Rewrite the text in the style of an editorial, making sure to include a clear stance or opinion and a concluding paragraph that calls for action or reflection. |
| 14 | Rewrite the text as an informational description, ensuring the tone is neutral and objective, and include at least one definition or clarification to help the reader better understand the subject. |
| 15 | Rewrite the text in the style of an encyclopedia entry, maintaining a neutral, authoritative tone, and include at least one date, fact, or reference to give it the appearance of being sourced. |
| 16 | Rewrite the text as an academic research article, structured with sections such as Abstract, Introduction, Method, Results, and Conclusion, and include at least one in-text citation (invented if necessary) to simulate scholarly referencing. |
| 17 | Rewrite the text as a descriptive profile of a specific thing or person, using vivid details and attributes (appearance, characteristics, or context) and ending with a short summary sentence that highlights its significance. |
| 18 | Rewrite the text in the form of a Frequently Asked Questions (FAQ) section, making sure to include at least three question–answer pairs, with the questions phrased from the perspective of a curious reader. |
| 19 | Rewrite the text as legal terms and conditions, using formal legal language, and ensure at least one numbered clause is included for clarity. |
| 20 | Rewrite the text as a personal opinion piece, written in the first person, making sure to clearly express a stance and support it with at least one reason or example. |
| 21 | Rewrite the text as a review, giving it a clear positive or negative stance, and include at least one specific detail or example to justify the evaluation. |
| 22 | Rewrite the text as an opinion blog post, written in a conversational and persuasive tone, and include at least one personal anecdote or illustrative example to strengthen the argument. |
| 23 | Rewrite the text as a denominational religious sermon, using a reverent and exhortative tone, and include at least one scriptural quotation or moral teaching to guide the audience toward reflection or action. |

Table 11: Outside register prompts

| ID | Synthesize Prompts |
|----|--------------------|
| 1 | Rewrite the following content as slide presentation bullet points. Focus on summarizing the key arguments and findings clearly and concisely. Use concise phrases that highlight core points. |
| 2 | Rewrite the following text in the style of a Facebook post. Sharing interesting information with followers. You may add light commentary, questions to the audience, or casual phrasing, but keep it natural and human-like. Avoid using emojis, hashtags, or overly dramatic expressions. |
| 3 | Adapt the text into a poetic form with vivid metaphors, rhythmic structure, and emotionally evocative language. |
| 4 | Convert the content into a tutorial-style explanation for beginners, using step-by-step instructions, simple analogies, and common misunderstandings. |
| 5 | Rewrite the text as a formal business email, ensuring clarity, professionalism, and a polite tone. |
| 6 | Rewrite the passage as a scientific abstract, including Background, Methods, Results, and Conclusions. Invent at least two numerical values (percentages, sample sizes, or statistical outcomes) to support claims. |
| 7 | Rewrite the text as a product description for an e-commerce website, highlighting key features, benefits, and use cases in a persuasive manner. |
| 8 | Rewrite the text as a blog post, incorporating vivid descriptions of locations, cultural insights, and personal experiences to engage readers. |
| 9 | Rewrite the text as a classroom lecture transcript, with explanations, rhetorical questions, and occasional student interaction. |
| 10 | Rewrite the text with stronger transitions between sentences and paragraphs, ensuring smoother reading without adding new information. |

though this instruction lacks a clear goal keyword, **SDR** infers a broader editorial-style prompt: "Rewrite the text in the style of an editorial, focusing on enhancing the narrative through emotional engagement, historical significance, and the subject's impact, while highlighting community involvement and contemporary relevance." Despite the absence of an explicit goal, this inferred prompt enhances the performance of unartificialized training data detection, particularly improving the AUC of Loss (Yeom et al., 2018) from 0.538 to 0.669.

## K    LIMITATION

A key limitation of our current approach is that the register taxonomy it relies on is too coarse-grained to locate goals accurately. Although the existing taxonomy of 23 sub-registers offers broad coverage of textual styles, it was not initially designed for classifying laundering goal. Consequently, there are cases where none of the 23 registers can adequately capture the intent of a synthesized prompt, leading to reduced accuracy in goal identification and, in turn, lower quality in restored prompts. Overcoming this limitation calls for future research on developing more fine-grained taxonomies tailored to synthesized data, thereby enabling more accurate and robust prompt reversal in practical scenarios.

## L    AI USAGE CLARIFICATION

Large Language Models improved the manuscript's grammar and readability; all research design, analysis, and interpretation were conducted by the authors.

## M    NEGATIVE-CONTROL EXPERIMENT

In the negative-control experiment setup where neither $D_{\text{por}}$ nor its laundered variants were used **SDR**. The result shows that **SDR** fails to find any prompt that improves detection: across all detectors,

Table 12: Inside register prompts and corresponding reversed prompts with Orig and `SDR` results.

| ID | Original Prompt | Reversed Prompt (`SDR`) | Orig | SDR |
|---|---|---|---|---|
| 1 | Rewrite the text in a lyrical style, ensuring the imagery is vivid, the rhythm flows naturally. | Rewrite the text in a lyrical style, enhancing the poetic rhythm and imagery while capturing the essence and emotional depth of the original content. | 0.540 | 0.692 |
| 2 | Rewrite the text in a spoken style, making it sound natural and conversational, and ensure the tone feels engaging and easy to follow for a live audience. | Rewrite the text to sound natural and conversational, using everyday language and personal anecdotes to create an engaging and friendly atmosphere for the listener. | 0.702 | 0.894 |
| 3 | Rewrite the text in the form of an interview, ensuring the questions flow naturally and the answers provide clear, engaging explanations for the audience. | Rewrite the text in the form of an interview, ensuring a clear and engaging dialogue that accurately conveys the information while maintaining a conversational tone and eliciting detailed responses. | 0.713 | 0.823 |
| 4 | Rewrite the text as an interactive discussion between two or more participants, ensuring the conversation flows logically, with each speaker's tone and style clearly distinguishable. | Rewrite the text as an interactive discussion between two or more participants, ensuring a natural flow of dialogue that incorporates factual information and engages with the topic through building on each other's comments. | 0.650 | 0.770 |
| 5 | Rewrite the text as a storytelling narrative. The story should flow naturally, use simple and engaging language, and be easy for all kinds of listeners to follow. | Rewrite the text in the form of a Frequently Asked Questions section, transforming the information into clear and concise questions and answers that emphasize key details and engage the reader effectively. | 0.646 | 0.735 |
| 6 | Rewrite the text in the style of a news report, ensuring the information is presented objectively and concisely. | Rewrite the text into a Frequently Asked Questions section, organizing the information into clear and concise questions and answers while highlighting key details and maintaining clarity and readability. | 0.793 | 0.904 |
| 7 | Rewrite the text as a sports report, ensuring the action is described with dynamic, energetic language that conveys the pace, tension, and excitement of the event. | Rewrite the text in the style of an engaging editorial, enhancing the narrative through vivid language, emotional depth, and a focus on the significance of the subject matter. | 0.655 | 0.713 |
| 8 | Rewrite the text as a narrative blog post, organized into clear sections with subheadings. Use a tone that is engaging and reflective, blending storytelling with explanation. | Rewrite the text in the form of a Frequently Asked Questions section, focusing on clearly structured questions and answers that highlight key aspects, contributions, and significance of the subject matter in a conversational tone. | 0.667 | 0.748 |
| 9 | Rewrite the text as a step-by-step instructional guide. Break the content into numbered steps, with each step beginning with a clear imperative verb. | Rewrite the text as a step-by-step instructional guide, breaking down the information into clear, organized steps that highlight key concepts, details, and relevant insights for enhanced understanding. | 0.778 | 0.841 |
| 10 | Rewrite the text as a recipe, introduce the information as sequential steps. | Rewrite the text to persuasively present factual information, emphasizing key aspects and structuring the content clearly to enhance engagement and clarity. | 0.700 | 0.724 |
| 11 | Rewrite the text to persuade the reader through factual information, making sure to include at least three specific data points or statistics to support the argument. | Rewrite the text in the style of an encyclopedia entry, focusing on enhancing structural clarity, coherence, and technical detail by organizing the information into distinct sections and emphasizing historical significance and key contributions. | 0.633 | 0.633 |
| 12 | Rewrite the text as a sales description, and be sure to include a clear call-to-action at the end. | Rewrite the text as a Frequently Asked Questions section, transforming the original content into a clear and engaging question-and-answer format that effectively highlights key elements, significance, and context for the reader. | 0.622 | 0.725 |

Table 13: Inside register prompts and corresponding reversed prompts with Orig and `SDR` results (continues).

| ID | Original Prompt | Reversed Prompt (`SDR`) | Orig | SDR |
|---|---|---|---|---|
| 13 | Rewrite the text in the style of an editorial, making sure to include a clear stance or opinion and a concluding paragraph that calls for action or reflection. | Rewrite the text as a personal opinion piece, emphasizing reflective commentary and personal insights while exploring the broader societal implications and significance of the subject matter. | 0.594 | 0.657 |
| 14 | Rewrite the text as an informational description, ensuring the tone is neutral and objective, and include at least one definition or clarification to help the reader better understand the subject. | Rewrite the text as a step-by-step instructional guide, organizing the content into clear, numbered sections that effectively communicate essential information about the subject. | 0.594 | 0.795 |
| 15 | Rewrite the text in the style of an encyclopedia entry, maintaining a neutral, authoritative tone, and include at least one date, fact, or reference to give it the appearance of being sourced. | Rewrite the text as an informational description, focusing on presenting a clear, structured overview of the subject's key facts, achievements, and background while maintaining concise and objective language. | 0.792 | 0.923 |
| 16 | Rewrite the text as an academic research article, structured with sections such as Abstract, Introduction, Method, Results, and Conclusion, and include at least one in-text citation (invented if necessary) to simulate scholarly referencing. | Rewrite the text in the form of an interview, transforming the original content into a conversational dialogue that incorporates engaging questions and responses while maintaining clarity and coherence. | 0.525 | 0.546 |
| 17 | Rewrite the text as a descriptive profile of a specific thing or person, using vivid details and attributes (appearance, characteristics, or context) and ending with a short summary sentence that highlights its significance. | Rewrite the text as a sales description, transforming it into an engaging narrative that highlights the subject's achievements, legacy, and emotional impact to captivate and appeal to potential audiences. | 0.661 | 0.765 |
| 18 | Rewrite the text in the form of a Frequently Asked Questions (FAQ) section, making sure to include at least three question–answer pairs, with the questions phrased from the perspective of a curious reader. | Rewrite the text in the form of an interview, transforming factual information into a conversational question-and-answer format that captures personal insights, key themes, and details from the original content. | 0.640 | 0.792 |
| 19 | Rewrite the text as legal terms and conditions, using formal legal language, and ensure at least one numbered clause is included for clarity. | Rewrite the text into a Frequently Asked Questions section by converting the content into clear questions and answers, ensuring clarity, conciseness, accuracy, and structured organization of information. | 0.731 | 0.833 |
| 20 | Rewrite the text as a personal opinion piece, written in the first person, making sure to clearly express a stance and support it with at least one reason or example. | Rewrite the text as a review, focusing on summarizing key aspects and implications while maintaining an engaging narrative style that connects with the reader. | 0.575 | 0.613 |
| 21 | Rewrite the text as a review, giving it a clear positive or negative stance, and include at least one specific detail or example to justify the evaluation. | Rewrite the text as a review, emphasizing the subject's significance, key achievements, and connections to broader themes or contexts, while maintaining a consistent tone and providing a balanced evaluation. | 0.593 | 0.698 |
| 22 | Rewrite the text as an opinion blog post, written in a conversational and persuasive tone, and include at least one personal anecdote or illustrative example to strengthen the argument. | Rewrite the text as a conversational interview, focusing on transforming factual content into dialogue by incorporating questions, responses, and personal insights while maintaining the original essence. | 0.577 | 0.630 |
| 23 | Rewrite the text as a denominational religious sermon, using a reverent and exhortative tone, and include at least one scriptural quotation or moral teaching to guide the audience toward reflection or action. | Rewrite the text as a denominational religious sermon, transforming the narrative into an inspirational message that emphasizes spiritual themes, fosters community, and resonates with the congregation's values. | 0.541 | 0.627 |

Table 14: Outside register prompts and corresponding reversed prompts with Orig and **SDR** results.

| ID | Original Prompt | Reversed Prompt (SDR) | Orig | SDR |
|---|---|---|---|---|
| 1 | Rewrite the following content as slide presentation bullet points. Focus on summarizing the key arguments and findings clearly and concisely. Use concise phrases that highlight core points. | Rewrite the text as a step-by-step instructional guide, organizing the information into clear sections and ensuring each step provides concise, relevant details on the specified topic. | 0.730 | 0.799 |
| 2 | Rewrite the following text in the style of a Facebook post. Sharing interesting information with followers. You may add light commentary, questions to the audience, or casual phrasing, but keep it natural and human-like. Avoid using emojis, hashtags, or overly dramatic expressions. | Rewrite the text as a Frequently Asked Questions section, transforming the information into an engaging question-and-answer format that encourages reader interaction and maintains a conversational tone. | 0.644 | 0.730 |
| 3 | Adapt the text into a poetic form with vivid metaphors, rhythmic structure, and emotionally evocative language. | Rewrite the text in a lyrical style that transforms factual content into an evocative narrative, using vivid imagery, poetic devices, and rhythmic flow to highlight emotional resonance and thematic cohesion. | 0.538 | 0.674 |
| 4 | Convert the content into a tutorial-style explanation for beginners, using step-by-step instructions, simple analogies, and common misunderstandings. | Rewrite the text as a step-by-step instructional guide, ensuring clear and concise steps that effectively outline key aspects and concepts while maintaining an engaging tone and logical flow throughout. | 0.616 | 0.679 |
| 5 | Rewrite the text as a formal business email, ensuring clarity, professionalism, and a polite tone. | Rewrite the text in the style of an encyclopedia entry, emphasizing clear and concise organization, formal language, and distinct sections that present factual information and key points effectively. | 0.795 | 0.884 |
| 6 | Rewrite the passage as a scientific abstract, including Background, Methods, Results, and Conclusions. Invent at least two numerical values (percentages, sample sizes, or statistical outcomes) to support claims. | Rewrite the text as a Frequently Asked Questions section, transforming the original content into clear, concise questions and answers that emphasize key themes, significant information, and factual accuracy. | 0.600 | 0.678 |
| 7 | Rewrite the text as a product description for an e-commerce website, highlighting key features, benefits, and use cases in a persuasive manner. | Rewrite the text as a dialogue in an interview format, emphasizing key details and insights while maintaining clarity and engagement through a question-and-answer structure. | 0.603 | 0.673 |
| 8 | Rewrite the text as a blog post, incorporating vivid descriptions of locations, cultural insights, and personal experiences to engage readers. | Rewrite the text as a sales description that emphasizes unique aspects and engaging narratives, highlighting significance and emotional appeal to captivate the audience. | 0.626 | 0.728 |
| 9 | Rewrite the text as a classroom lecture transcript, with explanations, rhetorical questions, and occasional student interaction. | Rewrite the text in the form of an interview, transforming the information into a natural dialogue that incorporates questions and answers while preserving the original content's key details and themes. | 0.667 | 0.764 |
| 10 | Rewrite the text with stronger transitions between sentences and paragraphs, ensuring smoother reading without adding new information. | Rewrite the text in the style of an editorial, focusing on enhancing the narrative through emotional engagement, historical significance, and the subject's impact, while highlighting community involvement and contemporary relevance. | 0.570 | 0.646 |

both AUC and ASR remain close to 0.5 on ArXiv and Wikipedia. This demonstrates that **SDR** does not spuriously create evidence of data misuse when $D_{\mathrm{pro}}$ was not used for training.

| Method | ArXiv | | Wiki | |
|---|---|---|---|---|
| | AUC | ASR | AUC | ASR |
| RecaLL | 0.516 | 0.532 | 0.528 | 0.545 |
| RecaLL + **SDR** | 0.488 | 0.525 | 0.473 | 0.540 |
| Loss | 0.486 | 0.522 | 0.471 | 0.510 |
| Loss + **SDR** | 0.490 | 0.535 | 0.465 | 0.515 |
| Min-K | 0.536 | 0.510 | 0.486 | 0.515 |
| Min-K + **SDR** | 0.499 | 0.535 | 0.440 | 0.525 |
| Min-K++ | 0.476 | 0.521 | 0.483 | 0.535 |
| Min-K++ + **SDR** | 0.483 | 0.520 | 0.446 | 0.515 |

Table 15: Negative-control experiment on **Pythia-6.9B**. When the target model is *not* trained on any laundered variants of the proprietary data, **SDR** does not find a prompt to improve unauthorized training-data detection. Across all detectors and both datasets (ArXiv and Wikipedia), the AUC and ASR scores before and after **SDR** remain close to 0.5, indicating that **SDR** does not fabricate evidence of misuse in the absence of laundered training data.

## N   SDR TRANSFERABILITY TO THIRD-PARTY LAUNDERING PIPELINES

**Alternative LLM Launderer with DeepSeek.** We fine-tune the target LLM exclusively on data laundered by DeepSeek-v3 Liu et al. (2024a), while **SDR** continues to use GPT-4o as the auxiliary model $M_a$ for prompt search. We apply the first five inside-register prompts (Table 10) as laundering templates and execute laundering with DeepSeek-v3. As shown in Table 16, **SDR** consistently improves detection performance across all four off-the-shelf detectors. For example, Loss improves from AUC/ASR of $0.65/0.64$ to $0.81/0.76$, and Min-K improves from $0.67/0.65$ to $0.78/0.73$. This demonstrates that **SDR** can recover useful transformations even when the laundering pipeline is completely different from GPT-4o.

| Method | AUC | ASR |
|---|---|---|
| RecaLL | 0.641 | 0.630 |
| RecaLL + **SDR** | **0.767** | **0.718** |
| Loss | 0.650 | 0.635 |
| Loss + **SDR** | **0.808** | **0.760** |
| Min-K | 0.672 | 0.645 |
| Min-K + **SDR** | **0.784** | **0.734** |
| Min-K++ | 0.555 | 0.584 |
| Min-K++ + **SDR** | **0.565** | **0.595** |

Table 16: **SDR** transferability on DeepSeek-v3 laundering pipelines for the target LLM trained exclusively on DeepSeek-laundered data. **SDR** uses GPT-4o as $M_a$.

**Human as Launderer (Polite Dataset).** While LLMs enable automated, large-scale rewriting practices, real-world laundering may not be carried out solely by LLMs. We simulate a laundering scenario using the Polite dataset (Wang et al., 2022), in which human annotators rewrote impolite sentences into polite versions. In this experiment, we treat the polite versions as "laundered" training data $D_{\mathrm{train}}$, while their original versions are treated as $D_{\mathrm{pro}}$. We find that **SDR** recovers a "personal review" style prompt (as shown below) and improves most off-the-shelf detectors (from Table 17).

We notice that RecaLL+**SDR** becomes slightly worse than RecaLL alone, which we attribute to polite expressions such as "I think" being already common in pretraining, making both training and non-training data equally easy to continue and collapsing the gap RecaLL relies on. Other detectors (+**SDR**) remain generally effective. In practice, Loss(+**SDR**) and (Min-K+**SDR**) may provide more reliable signals in this setting. Overall, these results demonstrate that **SDR** remains effective when the laundering pipeline uses a different LLM or human rewriting, and when the auxiliary and

| Method | AUC | ASR |
|---|---|---|
| RecaLL | **0.6488** | **0.635** |
| RecaLL + **SDR** | 0.5664 | 0.575 |
| Loss | 0.6866 | 0.680 |
| Loss + **SDR** | **0.7506** | **0.725** |
| Min-K | 0.6517 | 0.680 |
| Min-K + **SDR** | **0.7125** | **0.690** |
| Min-K++ | 0.5598 | 0.575 |
| Min-K++ + **SDR** | **0.6241** | **0.615** |

Table 17: Detection performance on the Polite (human-rewritten) dataset. **SDR** improves most detectors even when laundering is performed by humans, demonstrating **SDR**'s robustness and transferability beyond LLM-based pipelines.

laundering models are mismatched, confirming the transferability of **SDR** in diverse and practical scenarios.

## O  MIXED-REGISTER LAUNDERING PIPELINES

We have conducted experiments on **mixed-register transformations** by combining two registers in the laundering prompt (e.g., "opinion blog post with a persuasive tone", "storytelling narrative as a sports report", "informational description like a recipe"). Across all three mixed prompts, **SDR** consistently improves detection. For each mixed prompt, we present the recovered prompt followed by the corresponding detection results.

**Mixed-register prompt 1.**  Rewrite the text as an **opinion blog post** with a **persuasive tone**. **SDR**-*reversed prompt:* Rewrite the text as an opinion blog post, emphasizing personal narratives and reflections that highlight the emotional, cultural, or historical significance of the subject while engaging the reader.

**Mixed-register prompt 2.**  Rewrite the text as a **storytelling narrative** to introduce information as a **sports report**. **SDR**-*reversed prompt:* Rewrite the text as a narrative blog post. Focus on transforming the original information into a more engaging story, emphasizing the historical context and the evolution of the organization while maintaining a conversational tone.

**Mixed-register prompt 3.**  Rewrite the text as an **informational description** like a **recipe**. **SDR**-*reversed prompt:* Rewrite the text as a step-by-step instructional guide, clearly outlining key aspects and maintaining a logical flow and numbered steps.

| Method | Mixed Prompt 1 | | Mixed Prompt 2 | | Mixed Prompt 3 | |
|---|---|---|---|---|---|---|
| | AUC | ASR | AUC | ASR | AUC | ASR |
| RecaLL | 0.559 | 0.560 | 0.6656 | 0.650 | 0.7365 | 0.715 |
| RecaLL + **SDR** | **0.757** | **0.705** | **0.777** | **0.725** | **0.8313** | **0.770** |
| Loss | 0.572 | 0.575 | 0.6697 | 0.655 | 0.7375 | 0.685 |
| Loss + **SDR** | **0.767** | **0.705** | **0.796** | **0.750** | **0.834** | **0.785** |
| Min-K | 0.587 | 0.580 | 0.6767 | 0.670 | 0.7472 | 0.705 |
| Min-K + **SDR** | **0.750** | **0.715** | **0.749** | **0.700** | **0.820** | **0.765** |
| Min-K++ | 0.475 | 0.535 | 0.4749 | 0.515 | 0.5863 | 0.605 |
| Min-K++ + **SDR** | **0.586** | **0.615** | **0.538** | **0.580** | **0.617** | **0.615** |

Table 18: **SDR** transferability on mixed-register laundering prompts. **SDR** consistently improves AUC and ASR across three mixed-register laundering transformations, demonstrating robustness to more complex combinations of stylistic rewrites.

Across all three mixed-register prompts, **SDR** consistently improves both AUC and ASR, demonstrating that it can successfully transfer to mixed laundering transformations. Nevertheless, we

agree that extremely exotic transformations (e.g., pseudo-translation into low-resource languages) may further stress **SDR**.

## P    HYPERPARAMETER SENSITIVITY STUDY.

We conducted an extended sensitivity study to examine how **SDR** behaves under different choices of the key hyperparameters $K$ (see Table 19), $l$ (see Table 20), $m$ (see Table 21), and $n$ (see Table 22), using `Loss`+**SDR** under the No. 1 inside-register prompt.

Our key findings are as follows (see the tables below for detailed numbers): Increasing $K$ from $3 \to 15$ improves AUC from $0.71 \to 0.75$ and ASR from $0.69 \to 0.71$, at the cost of higher runtime and query budget (roughly \$7–\$39 with GPT-4o; substantially cheaper with GPT-4o-mini). The value of $m$ has the strongest impact. Varying $m$ from $3 \to 9$ raises AUC from $0.72 \to 0.81$ and ASR from $0.69 \to 0.76$ with almost unchanged query cost, making $m = 9$ a strong default choice. $l$ and $n$ exhibit moderate gains. Setting $l = 7$ and $n = 7$ achieves a good balance between performance and cost. *Note that $n$ is used only once when building templates and does not affect per-run latency.*

**Detailed numerical results.** Unless otherwise specified, we use $K = 10, l = 5, m = 5$, and $n = 10$ as the default hyperparameters.

Table 19: Sensitivity to $K$

| $K$ | AUC | ASR | Time | Query Budget (GPT4o / mini) |
|---|---|---|---|---|
| 3 | 0.712 | 0.685 | 02:23:42 | $\sim \$7/0.8$ |
| 5 | 0.727 | 0.690 | 04:11:33 | $\sim \$13/1.5$ |
| 15 | 0.747 | 0.705 | 13:27:12 | $\sim \$39/4.5$ |

Table 20: Sensitivity to $l$

| $l$ | AUC | ASR | Time | Query Budget (GPT4o / mini) |
|---|---|---|---|---|
| 3 | 0.715 | 0.670 | 08:21:33 | $\sim \$10/1$ |
| 7 | 0.731 | 0.695 | 09:13:12 | $\sim \$10/1$ |
| 9 | 0.718 | 0.695 | 09:35:17 | $\sim \$10/1$ |

Table 21: Sensitivity to $m$

| $m$ | AUC | ASR | Time | Query Budget (GPT4o / mini) |
|---|---|---|---|---|
| 3 | 0.718 | 0.685 | 08:14:33 | $\sim \$10/1$ |
| 7 | 0.796 | 0.740 | 08:19:29 | $\sim \$10/1$ |
| 9 | 0.810 | 0.760 | 08:26:45 | $\sim \$10/1$ |
| 11 | 0.806 | 0.755 | 08:33:34 | $\sim \$10/1$ |

Overall, a practical and efficient configuration for **SDR** is: $(K = 5, \quad l = 7, \quad m = 9, \quad n = 7)$, which attains most of the performance gains at moderate cost. We will include these tables and a concise discussion in the Appendix.

## Q    COMPARISON TO REVERSE PROMPT ENGINEERING METHODS

To clarify how **SDR** differs from related lines of work, especially those based on prompt-recovery techniques, we note an important methodological distinction: To the best of our knowledge, existing *reverse prompt engineering* methods, however, assume that the unknown prompt is applied at *inference* time and leaves a direct trace in the observed outputs. In our setting, the laundering prompt is

Table 22: Sensitivity to $n$. $n$ controls the construction of the template and is computed only once during initialization. Therefore, it does not contribute to the runtime or query budget.

| $n$ | AUC | ASR | Time | Query Budget |
|---|---|---|---|---|
| 3 | 0.705 | 0.675 | – | – |
| 7 | 0.752 | 0.705 | – | – |
| 9 | 0.742 | 0.695 | – | – |

applied before *training*. The data-rights holder only sees the trained model and never observes any text generated under the unknown laundering prompt, making a direct application impossible.

We summarize the conceptual differences between Reverse Prompt Engineering (Li & Klabjan, 2024) and `SDR` as follows. Reverse prompt engineering assumes that the unknown prompt is applied during inference, meaning that it directly conditions the model's outputs and therefore leaves observable traces that the recovery procedure can exploit. In contrast, `SDR` targets laundering transformations applied before training, where the prompt does not manifest in any generated text. Instead, only implicit consequences of the laundering process remain embedded in the trained model. Moreover, reverse prompt engineering methods rely on having direct access to model outputs produced under the unknown prompt, whereas in our setting the data-rights holder never observes any prompt-conditioned outputs and can only query the final trained model. These differences make existing reverse prompt engineering approaches incompatible with the laundering-before-training scenario that SDR is designed to address.

To still provide a comparison, we slightly adapt Reverse Prompt Engineering (Li & Klabjan, 2024) to a use case better aligned with our problem setup: given an original document, we let $M_t$ generate a continuation and treat that continuation as a proxy "laundered" version; we then ask Reverse Prompt Engineering (Li & Klabjan, 2024) to infer a prompt connecting the original and continuation.

Table 23 shows that `SDR` provides the largest improvement among all methods. `Loss`+`SDR` achieves an AUC of 0.782 and an ASR of 0.764, clearly outperforming both the baseline detector and the adapted reverse prompt engineering approach. While reverse prompt engineering offers modest gains over the baseline, its improvements remain limited, highlighting that it is considerably less effective than `SDR` in recovering laundering transformations and restoring detector performance.

Table 23: Comparison between SDR and reverse prompt engineering.

| Method | AUC | ASR |
|---|---|---|
| **Loss + SDR** | **0.782** | **0.764** |
| Loss + Reverse Prompt Engineering | 0.682 | 0.663 |
| Loss | 0.638 | 0.635 |

## R   TPR@1FPR RESULT

We additionally report the TPR@1% results for the main experiment.

Table 24: TPR@1% results for the main experiment.

| Method | Inside TPR@1 | Outside TPR@1 |
|---|---|---|
| Recall | 0.002 | 0.008 |
| Recall + **SDR** | **0.106** | **0.062** |
| Loss | 0.047 | 0 |
| Loss + **SDR** | **0.112** | **0.069** |
| Min-K | 0.0013 | 0.016 |
| Min-K + **SDR** | **0.096** | **0.082** |
| Min-K++ | 0.002 | 0.012 |
| Min-K++ + **SDR** | **0.086** | **0.071** |

## S  PARTIAL LAUNDERING EXPERIMENT

We conducted an experiment to evaluate whether SDR remains effective when only a portion of the proprietary data is laundered. Given $|D_{\text{pro}}| = 200$, we randomly selected half of the samples and applied the first five inside-register prompts (Appendix Table 9) to generate the laundered subsets used for training, while the remaining samples were left unaltered. SDR was then applied to the full $D_{\text{pro}}$ to infer the laundering transformation. The table below reports the average performance across all five prompts.

Table 25: Average performance across all five laundering prompts under partial laundering.

| Method | Avg AUC | Avg ASR |
|---|---|---|
| RecaLL | 0.608 | 0.590 |
| RecaLL + SDR | **0.722** | **0.694** |
| Loss | 0.599 | 0.597 |
| Loss + SDR | **0.732** | **0.703** |
| Min-K | 0.608 | 0.603 |
| Min-K + SDR | **0.724** | **0.701** |
| Min-K++ | 0.553 | 0.565 |
| Min-K++ + SDR | **0.580** | **0.602** |

SDR successfully improves detection performance across all four off-the-shelf detectors. For example, the AUC/ASR of Loss increases from 0.599/0.597 to 0.732/0.703.

