# OpenReview forum: "Combating Data Laundering in LLM Training"
_ICLR.cc/2026/Conference — Submitted to ICLR 2026_

### Official Review · Reviewer_GhSm · 2025-10-20

**Soundness:** 2
**Presentation:** 3
**Contribution:** 2
**Rating:** 6
**Confidence:** 3

**Summary:**

The paper addresses the challenge of detecting unauthorized data use in large language model (LLM) training when proprietary data has been "laundered" through semantic-preserving transformations (e.g., stylistic rewriting) to evade detection. The authors propose Synthesis Data Reversion (SDR), a two-stage method that infers the unknown laundering transformation from black-box access to the target LLM using an auxiliary LLM. SDR abstracts transformations into a high-level goal (e.g., "lyrical rewriting") from a taxonomy of 23 linguistic registers and refines it with iterative details to synthesize "training-like" queries. Evaluations on the MIMIR benchmark across LLM families (Pythia, Llama-2, Falcon) show SDR consistently improves detection metrics under various laundering scenarios.

**Strengths:**

1. It is timely and important to investigate the data laundering in LLM training
2. Strong detection performance compared with baseline methods.

**Weaknesses:**

1. The proposed method Involves multiple iterative queries to both target and auxiliary LLMs, potentially costly for large datasets or real-world audits, with no detailed cost analysis provided.
2. It would be better if the authors could report more results via TPR@1%, considering such an audit task requires much confidence.
3. There is another concern about the assumption, the authors assumes laundering follows a promptable goal-details structure executable by an auxiliary LLM, however, real-world laundering might be more opaque or non-prompt-based (e.g., human-edited).

**Questions:**

1. What is the specific computational cost of the proposed method?
2. Can the proposed method effectively address real-world laundering (e.g., human-edited)?

---

> ### Author Response · Authors · 2025-11-23
>
> We thank the reviewer for the thoughtful comments on computational cost, high‑confidence metrics, and the realism of the goal–details assumption. Below, we provide a concrete cost analysis, clarify our use of TPR@1%, and add new experiments on both third‑party LLM and human‑edited laundering pipelines.
>
> > **W1.** The proposed method involves multiple iterative queries to both target and auxiliary LLMs, potentially costly for large datasets or real-world audits, with no detailed cost analysis provided. What is the specific computational cost of the proposed method?
>
> > **Q1.** What is the specific computational cost of the proposed method?
>
>
> #### What is the specific computational cost of the proposed method?
> We have conducted sensitivity study on how the main SDR hyperparameters (e.g., search depth $l$, register candidates $m$, and generation rounds $K$) affect resource consumption and performance, using Loss+SDR on MIMIR‑Wikipedia with $|D_{\rm pro}| = 100$ and $|D_{\rm held}| = 100$ under inside‑register prompt No.1.
>
> Our key findings are (see the tables below for detailed numbers):
> - increasing $K$ from $3 \to 15$ improves AUC from 0.71 to 0.75 and ASR from 0.69 to 0.71, at the cost of higher runtime and query budget (roughly \$7-\$39 with GPT-4o, substantially cheaper with GPT-4o-mini)
> - The value of $m$ has the strongest impact. Changing $m$ between $3 \to 9$ raises AUC from 0.72 to 0.81 and ASR from 0.69 to 0.76 with almost unchanged query cost, making $m =9$ a reasonable default.
> - $l$ and $n$ exhibit moderate improvements. Setting $l=7$ and $n=7$ balances performance and cost. *Note that $n$ is used only once when building templates and does not affect per‑run latency.*
>
>
> #### Detailed numerical results on hyperparameter sensitivity
> We use $K$=10, $l$=5, $m$=5, and $n$=10 as the default hyperparameters unless otherwise specified.
> | $K$ | AUC   | ASR   | Time    | Query Budget (GPT4o/GPT4omini) |
> |---|-------|-------|-----------|--------------|
> | 3 | 0.712 | 0.685 | 02:23:42  | ~\$7/$0.8         |
> | 5 | 0.727 | 0.690 | 04:11:33  | ~\$13/$1.5        |
> | 15| 0.747 | 0.705 | 13:27:12  | ~\$39/$4.5        |
>
> | $l$ | AUC   | ASR   | Time      | Query Budget (GPT4o/GPT4omini) |
> |---|-------|-------|-----------|--------------|
> | 3 | 0.715 | 0.670 | 08:21:33  | ~\$10/$1        |
> | 7 | 0.731 | 0.695 | 09:13:12  | ~\$10/$1         |
> | 9 | 0.718 | 0.695 | 09:35:17  | ~\$10/$1         |
>
> | $m$ | AUC   | ASR   | Time      | Query Budget (GPT4o/GPT4omini) |
> |---|-------|-------|-----------|--------------|
> | 3 | 0.718 | 0.685 | 08:14:33  | ~\$10/1         |
> | 7 | 0.796 | 0.740 | 08:19:29  | ~\$10/1         |
> | 9 | 0.810 | 0.760 | 08:26:45  | ~\$10/1         |
> | 11|0.806 | 0.755 | 08:33:34| ~\$10/1 |
>
> | $n$ | AUC   | ASR   | Time      | Query Budget (GPT4o/GPT4omini) |
> |---|-------|-------|-----------|--------------|
> | 3 | 0.705 | 0.675 | —         | —            |
> | 7 | 0.752 | 0.705 | —         | —            |
> | 9 | 0.742 | 0.695 | —         | —            |
>
> *Note: $n$ controls the construction of the template and is computed only once at initialization; therefore, it does not contribute to the runtime or query budget.*
>
> Thus, a practical configuration that *balances performance and cost* is $(K=5\, l=7\, m=9\, n=7)$, which we will highlight in the revision.
> Because each SDR iteration rewrites and evaluates each sample once, the total cost scales roughly linearly in the number of audited samples.
> In practice, the individual rights holder typically does not have a very large number of candidate documents to audit, and we believe the cost of SDR is acceptable at the moderate scale of proprietary samples.
>
> #### GPT-4o provides a more cost-effective alternative
> When it comes to the need to handle large-scale datasets, switching the auxiliary LLM $M_a$ to a more cost-efficient LLM (e.g., GPT-4o mini) provides a viable solution, which reduces costs by ~90% with only a marginal drop in AUC (see tables below for numerical results).
>
> Result on **GPT4o**:
> | K | AUC   | ASR   | Query Budget (GPT4o/GPT4omini) |
> |---|-------|-------|--------------|
> | 3 | 0.712 | 0.685 | ~\$7         |
>
> Result on **GPT4omini**:
> | K | AUC   | ASR      | Query Budget (GPT4o/GPT4omini) |
> |---|-------|-------|--------------|
> | 3 | 0.695 | 0.655   | ~$0.8      |
>
> This makes GPT-4o mini (and other small LLMs) a practical choice for scaling SDR to large datasets.

---

> > ### Author Response · Authors · 2025-11-23
> >
> > > **W2.** It would be better if the authors could report more results via TPR@1%, considering that such an audit task requires much confidence.
> >
> > We agree that reporting TPR@1% is important for high-confidence auditing scenarios.
> > Here, we additionally report the TPR@1% results for the main experiment (**Table 2** in the original paper).
> >
> > | Method        | Inside TPR@1 | Outside TPR@1 |
> > |---------------|--------------|----------------|
> > | Recall        | 0.002        | 0.008          |
> > | Recall + **SDR**  | **0.106**    | **0.062**      |
> > | Loss            | 0.047        | 0              |
> > | Loss + **SDR**      | **0.112**    | **0.069**      |
> > | Min-K         | 0.0013       | 0.016          |
> > | Min-K + **SDR**   | **0.096**    | **0.082**      |
> > | Min-K++       | 0.002        | 0.012          |
> > | Min-K++ + **SDR** | **0.086**    | **0.071**      |
> >
> >
> > In the revised version, we will include TPR@1% results for all major experiments in the Appendix to provide a stronger assessment of SDR’s effectiveness under strict false-positive constraints.

---

> ### Author Response · Authors · 2025-11-23
>
> > **W3.** There is another concern about the assumption; the authors assume laundering follows a promptable goal-details structure executable by an auxiliary LLM; however, real-world laundering might be more opaque or non-prompt-based (e.g., human-edited).
>
> > **Q2.** Can the proposed method effectively address real-world laundering (e.g., human-edited)?
>
> Thanks for raising this issue. We address the concern that real-world laundering may not follow a *single goal-details structure* by additionally evaluating SDR on two other forms of laundering pipelines, beyond the settings we reported in Sec. 5 and Table 5, where SDR already transfers across auxiliary LLMs (GPT‑4o, Claude, DeepSeek) with a fixed GPT‑4o laundering pipeline.
>
> #### **Human As Launderer (with Polite Dataset [1]).**
> We first evaluate *cross‑launderer* transfer with human-rewrite-based laundering.
>
> While LLMs enable automated, large-scale rewriting practices, real-world laundering may not be executed by LLMs solely. We simulate a laundering scenario using the Polite dataset [1], where human annotators rewrote impolite sentences to be polite versions.
>
> In this experiment, we treat the polite versions as "laundered" training data $D_{\rm train}$, while their original versions are treated as $D_{\rm pro}$. We find that SDR recovers a “personal review" style prompt (as shown below) and **improves most off-the-shelf detectors**.
>
> ***SDR-reversed prompt:** Rewrite the text as a personal review, emphasizing emotional responses, personal opinions, and experiences while maintaining a reflective and conversational tone.*
>
> **Table: Performance on the Polite (Human-Rewritten) Dataset**
>
> |Method|AUC|ASR|
> |-|-|-|
> |RecaLL|**0.6488**|**0.635**|
> |RecaLL+**SDR**|0.5664|0.575|
> |Loss|0.6866|0.680|
> |Loss+**SDR**|**0.7506**|**0.725**|
> |Min-K|0.6517|0.680|
> |Min-K+**SDR**|**0.7125**|**0.690**|
> |Min-K++|0.5598|0.575|
> |Min-K+++**SDR**|**0.6241**|**0.615**|
>
> We notice that RecaLL+SDR becomes slightly worse than RecaLL alone, which we attribute to polite expressions such as “I think” being already common in pretraining, making both training and non-training data equally easy to continue and collapsing the gap RecaLL relies on. Other detectors (+SDR) remain generally effective. In practice, Loss(+SDR) and (Min-K+SDR) may provide more reliable signals in this setting.

---

> ### Author Response · Authors · 2025-11-23
>
> **We continue our response from the previous section.**
>
> #### **Mixed-register Transformation**
> We further simulate a "more complex" transformation than the single goal-detail prompt structure when executing laundering with LLMs.
>
> We have now conducted experiments on **mixed-register transformations** by combining two registers in the laundering prompt (e.g., “opinion blog post with a persuasive tone”, “storytelling narrative as a sports report”, “informational description like a recipe”).
>
> Across three such mixed prompts, SDR consistently improves detection. For each mixed prompt, we present the corresponding prompt recovered by SDR, followed by tables reporting the performance changes of baseline detectors before and after applying SDR.
>
> ---
> - *Mixed-register prompt 1:* Rewrite the text as an **opinion blog post** with a **persuasive tone**.
>     - *Reversed prompt:* Rewrite the text as an opinion blog post, emphasizing personal narratives and reflections that highlight the emotional, cultural, or historical significance of the subject while engaging the reader.
>
> - *Mixed-register prompt 2:* Rewrite the text as a **storytelling narrative** to introduce information as a **sports report**.
>     - *Reversed prompt:* Rewrite the text as a narrative blog post. Focus on transforming the original information into a more engaging story, emphasizing the historical context and the evolution of the organization while maintaining a conversational tone.
>
> - *Mixed-register prompt 3:* Rewrite the text as an **informational description** like a **recipe**.
>     - *Reversed prompt:* Rewrite the text as a step-by-step instructional guide, clearly outlining key aspects and maintaining a logical flow and numbered steps.
>
> ---
> **Table: Detection performance on mixed-register prompt**
> | Method | Mixed Prompt 1 AUC | Mixed Prompt 1 ASR | Mixed Prompt 2 AUC | Mixed Prompt 2 ASR | Mixed Prompt 3 AUC | Mixed Prompt 3 ASR |
> |-|-|-|-|-|-|-|
> |Recall|0.559|0.560|0.6656|0.650|0.7365|0.715|
> |Recall+**SDR**|**0.757**|**0.705**|**0.777**|**0.725**|**0.8313**|**0.770**|
> |Loss|0.572|0.575|0.6697|0.655|0.7375|0.685|
> |Loss+**SDR**|**0.767**|**0.705**|**0.796**|**0.750**|**0.834**|**0.785**|
> |Min-K|0.587|0.580|0.6767|0.670|0.7472|0.705|
> |Min-K+**SDR**|**0.750**|**0.715**|**0.749**|**0.700**|**0.820**|**0.765**|
> |Min-K++|0.475|0.535|0.4749|0.515|0.5863|0.605|
> |Min-K++ +**SDR**|**0.586**|**0.615**|**0.538**|**0.580**|**0.617**|**0.615**|
> Across all three mixed-register prompts, SDR consistently improves both AUC and ASR, demonstrating that it can successfully transfer to mixed laundering transformations.
>
>
> #### Unseen-register transformation
> Beyond *mixed-register transformation*, it is also common that the data rights holder does not know possible domains that the model provider may have laundered $D_{\rm pro}$ into. This represents an "opaque" laundering setup in practice.
>
> To simulate such cases, we've evaluated **outside-register prompts**, i.e., prompts generated specifically not to align with any of the 23 registers (in Sec. 5). Across these "out-of-distribution" transformations, SDR still yields substantial AUC gains (Tables (2-3)).
>
>
> #### Summary
> Overall, these results demonstrate that SDR remains effective when the laundering pipeline is executed by human rewriting, and when the prompt-based laundering is complex, confirming the *transferability and robustness of SDR* in diverse and practical scenarios.
>
> Nevertheless, we agree that extremely exotic transformations (e.g., pseudo‑translation into low-resource languages) and more complicated laundering pipelines may further stress SDR, and we will revise Appendix I to highlight that more task‑specific taxonomies are an important direction for future work rather than a solved problem.
>
> ##### References
> ###### [1] Wang, Xun, et al. "Pay attention to your tone: Introducing a new dataset for polite language rewrite." arXiv preprint arXiv:2212.10190 (2022).
>
> ---
> We sincerely hope these clarifications address the reviewer’s concerns. If there are any remaining questions, we would be happy to address them. Otherwise, we would very appreciate if the reviewer could reconsider and raise the scores.

---

> > ### Comment · Reviewer_GhSm · 2025-11-28
> >
> > Thank you for your response. Most of my concerns have been addressed. Regarding W2, I note that the proposed algorithm does indeed exhibit a low TPR, which could hinder its future application. I would like to know if more results and analysis could be presented, or if you could explain how this issue might be improved in the future and how this method could be deployed effectively and reliably.

---

> > > ### Author Response · Authors · 2025-12-03
> > >
> > > Thank you very much for the follow-up and for raising this point regarding the low TPR@1. Although the absolute TPR@1% values (~6–11%) are modest, this is because the task introduced by SDR (detecting unauthorized training data in black-box LLMs under data laundering) is **difficult** for existing unauthorized training data detection methods.
> > >
> > > **Existing detection methods only identify training samples that appear exactly in the model’s training set and are not designed to detect content-preserved but style-altered data (e.g., laundered data).** When the target model is trained on laundered data, most original training samples are treated as non-training data because they do not match the laundered training data. Consequently, under the strict TPR@1% FPR setting, achieving a high absolute TPR is challenging.
> > >
> > > For example, when querying a Pythia-6.9B model trained on Wikipedia data luandered with the prompt “Rewrite the text as a recipe.”, ReCaLL obtains **0%** TPR@1% when evaluated on the original training and non-training samples [1].
> > >
> > > SDR is a modular framework that does not rely on a particular detector, and its performance depends on the capability of the underlying detector. While SDR can partially recover laundered data, current detectors may still classify these recovered samples as non-training data. Thus, the modest absolute TPR@1% reflects limitations of existing detectors in low-FPR regimes under data laundering, rather than limitations of SDR. Despite this, SDR provides at least a **240%** relative improvement over the baselines, allowing detectors to identify a subset of potentially misused data.
> > >
> > > In practical auditing scenarios, such levels of sensitivity can still be informative. The aim of unauthorized training-data detection is often to assess whether any unauthorized use occurred, rather than to identify all involved samples. For instance, a TPR@1% of around 10% implies that querying 1,000 proprietary samples yields roughly 100 detections at a 1% FPR operating point, which can offer evidence supporting further internal review or legal processes [2].
> > >
> > > As more capable detectors are developed, the improvements obtained through SDR are expected to translate into higher absolute detection accuracy. Future detection research may also consider treating semantically equivalent transformed data as potential training data.
> > >
> > > ### References
> > > [1] Xie, Roy, et al. "ReCaLL: Membership Inference via Relative Conditional Log-Likelihoods." EMNLP 2024.
> > >
> > > [2] Carlini, Nicholas, et al. "Membership inference attacks from first principles." IEEE S&P 2022.

---

> ### Author Response · Authors · 2025-11-27
> **Follow-up on rebuttal clarifications**
>
> Dear Reviewer GhSm,
>
> With the discussion period closing in less than 6 days, we are writing to kindly follow up on our rebuttal.
>
> We would greatly appreciate it if you could take a moment to review our response. If you find that these new results address your concerns, we would value the reconsideration of your assessment.
>
> We remain available to answer any further questions you might have.
>
> Best regards,
>
> The Authors

---

### Official Review · Reviewer_zZxJ · 2025-11-01

**Soundness:** 3
**Presentation:** 3
**Contribution:** 3
**Rating:** 4
**Confidence:** 3

**Summary:**

The paper tackles data laundering in LLM training, where copyrighted texts are stylistically transformed to evade provenance checks. It introduces Synthesis Data Reversion (SDR), a black box method that first infers a high level laundering goal (e.g., lyrical rewrite) and then refines concrete stylistic details, using an auxiliary LLM to generate probes that match the laundered style. These probes restore detection gaps so the target model again reveals training exposure. On the MIMIR benchmark across diverse laundering practices and model families, SDR consistently boosts misuse detection.

**Strengths:**

1. Clearly shows that standard post hoc provenance tests collapse when models are trained on laundered surrogates rather than originals, making detection with losses or calibrated confidence ineffective. The setup and failure case are well motivated and demonstrated.
2. Reframe detection as reversing the unknown laundering transform, then reuse off-the-shelf detectors. The two-stage SDR pipeline uses a goal, then details abstraction over registers to search a compact prompt space with only black box access. Algorithms 1 and 2 are clear and practical.
3. In the experiment section, SDR consistently boosts several detectors across prompts, datasets, and model families and works with different auxiliary LLMs.

**Weaknesses:**

1. Most results rely on the laundering produced by GPT-style rewriting under predefined prompts. And the GPT may introduce a new bias. How about adding a third-party laundering pipeline, which could strengthen the whole paper.
2. SDR needs repeated calls to an auxiliary model to build templates and iterate prompts. The experiment section only includes a limited introduction of query budgets, latency, and sensitivity to n, m, l, and K. A more comprehensive sensitivity study would help strengthen the paper.
3. Results compare SDR plus standard detectors to the detectors alone. Since the contribution is a laundering-aware search, comparisons to other data-centric countermeasures or prompt search strategies would help to improve.
4. The goal identification stage assumes the laundering transformation belongs to one of the 23 predefined registers. Though this is noted in Appendix I, many realistic transformations, like pseudo-translation or hybrid creative styles, etc, may limit SDR’s recall and generalization. Moreover, the method is not tested on unseen or mixed registers, leaving its robustness to out-of-distribution transformations uncertain.

**Questions:**

Please see the weakness section. I will raise the score if the author addresses the questions clearly

---

> ### Author Response · Authors · 2025-11-23
> **Response on model transferability clarification**
>
> # Response on model transferability clarification (W1 & W4)
>
> We thank the reviewer for raising the questions on transferability across laundering pipelines and on the coverage of the 23‑register taxonomy. We summarize new experiments and clarifications below.
>
> > **W1:** Most results rely on the laundering produced by GPT-style rewriting under predefined prompts. And the GPT may introduce a new bias. How about adding a third-party laundering pipeline, which could strengthen the whole paper?
>
> Besides the settings in Sec. 5 and Table 5, where SDR already transfers across auxiliary LLMs (GPT‑4o, Claude, DeepSeek) with a fixed GPT‑4o laundering pipeline, we now evaluate *cross‑launderer* transfer with two forms of third-party laundering pipelines.
>
> #### **1. Alternative LLM Launderer with DeepSeek.**
> We fine-tune the target LLM only on data laundered by **DeepSeek-v3**, while SDR continues to use GPT-4o as $M_a$ for prompt searching.
> <!-- *We train target models exclusively on **DeepSeek-laundered data**. SDR still using **GPT4o** to reverse the prompt.* -->
>
> We apply the first 5 inside-register prompts (Appendix Table 9) as the laundering prompts and execute the laundering using **DeepSeek-v3**.
> As shown below, SDR successfully improves detection performance across all four off-the-shelf detectors.
> For example, Loss’s AUC/ASR increase from 0.65/0.64 to 0.81/0.76 and Min‑K from 0.67/0.65 to 0.78/0.73.
>
> **Table: SDR transferability on DeepSeek-v3 laundering pipelines**
> | Method | AUC | ASR |
> |--------|-----|-----|
> | RecaLL | 0.641 | 0.630 |
> | RecaLL + **SDR** | **0.767** | **0.718** |
> | Loss | 0.650 | 0.635 |
> | Loss + **SDR** | **0.808** | **0.760** |
> | Min-K | 0.672 | 0.645 |
> | Min-K + **SDR** | **0.784** | **0.734** |
> | Min-K++ | 0.555 | 0.584 |
> | Min-K++ + **SDR** | **0.565** | **0.595** |
>
> This shows that SDR can recover useful transformations when the launderer is not GPT-4o.
>
>
> #### **2. Human As Launderer (with Polite Dataset [1]).**
>
> While LLMs enable automated, large-scale rewriting practices, real-world laundering may not be executed by LLMs solely. We simulate a laundering scenario using the Polite dataset [1], where human annotators rewrote impolite sentences to be polite versions.
>
> In this experiment, we treat the polite versions as "laundered" training data $D_{\rm train}$, while their original versions are treated as $D_{\rm pro}$. We find that SDR recovers a “personal review" style prompt (as shown below) and **improves most off-the-shelf detectors**.
>
> ***SDR-reversed prompt:** Rewrite the text as a personal review, emphasizing emotional responses, personal opinions, and experiences while maintaining a reflective and conversational tone.*
>
> **Table: Performance on the Polite (Human-Rewritten) Dataset**
>
> | Method        | AUC     | ASR     |
> |---------------|---------|---------|
> | RecaLL        | **0.6488**  | **0.635**   |
> | RecaLL + **SDR**  | 0.5664  | 0.575   |
> | Loss            | 0.6866  | 0.680   |
> | Loss + **SDR**      | **0.7506**  |**0.725**   |
> | Min-K         | 0.6517  | 0.680   |
> | Min-K + **SDR**   | **0.7125**  | **0.690**   |
> | Min-K++       | 0.5598  | 0.575   |
> | Min-K++ + **SDR** | **0.6241**  | **0.615**  |
>
> We notice that RecaLL+SDR becomes slightly worse than RecaLL alone, which we attribute to polite expressions such as “I think” being already common in pretraining, making both training and non-training data equally easy to continue and collapsing the gap RecaLL relies on. Other detectors (+SDR) remain generally effective. In practice, Loss(+SDR) and (Min-K+SDR) may provide more reliable signals in this setting.
>
> Overall, these results demonstrate that SDR remains effective when the laundering pipeline uses a different LLM or human rewriting, and when the auxiliary and laundering models are mismatched, confirming the *transferability of SDR* in diverse and practical scenarios.

---

> ### Author Response · Authors · 2025-11-23
>
> > **W4.** The goal identification stage assumes the laundering transformation belongs to one of the 23 predefined registers. Though this is noted in Appendix I, many realistic transformations, like pseudo-translation or hybrid creative styles, etc, may limit SDR’s recall and generalization. Moreover, the method is not tested on unseen or mixed registers, leaving its robustness to out-of-distribution transformations uncertain.
>
> Thank you for the valuable suggestion. We apologize that the robustness of SDR beyond the 23 registers was not sufficiently emphasized.
>
> #### **Unseen-register transformation**
> As noted in Sec. 5, we've already evaluated **outside-register prompts**, i.e., prompts generated specifically not to align with any of the 23 registers. Across these "out-of-distribution" transformations, SDR still yields substantial AUC gains (Tables (2-3)).
>
>
> #### **Mixed-register transformation**
> In addition, we have now conducted experiments on **mixed-register transformations** by combining two registers in the laundering prompt (e.g., “opinion blog post with a persuasive tone”, “storytelling narrative as a sports report”, “informational description like a recipe”).
>
> Across three such mixed prompts, SDR consistently improves detection. For each mixed prompt, we present the corresponding prompt recovered by SDR, followed by tables reporting the performance changes of baseline detectors before and after applying SDR.
>
> ---
> - *Mixed-register prompt 1:* Rewrite the text as an **opinion blog post** with a **persuasive tone**.
>     - *Reversed prompt:* Rewrite the text as an opinion blog post, emphasizing personal narratives and reflections that highlight the emotional, cultural, or historical significance of the subject while engaging the reader.
>
> - *Mixed-register prompt 2:* Rewrite the text as a **storytelling narrative** to introduce information as a **sports report**.
>     - *Reversed prompt:* Rewrite the text as a narrative blog post. Focus on transforming the original information into a more engaging story, emphasizing the historical context and the evolution of the organization while maintaining a conversational tone.
>
> - *Mixed-register prompt 3:* Rewrite the text as an **informational description** like a **recipe**.
>     - *Reversed prompt:* Rewrite the text as a step-by-step instructional guide, clearly outlining key aspects and maintaining a logical flow and numbered steps.
>
> ---
> **Table: Detection performance on mixed-register prompt**
> | Method            | Mixed Prompt 1 AUC | Mixed Prompt 1 ASR | Mixed Prompt 2 AUC | Mixed Prompt 2 ASR | Mixed Prompt 3 AUC | Mixed Prompt 3 ASR |
> |-------------------|--------------------|---------------------|---------------------|---------------------|---------------------|---------------------|
> | RecaLL            | 0.559              | 0.560               | 0.6656              | 0.650               | 0.7365              | 0.715               |
> | RecaLL + **SDR**      | **0.757**              | **0.705**               | **0.777**              | **0.725**               | **0.8313**              | **0.770**               |
> | Loss                | 0.572              | 0.575               | 0.6697              | 0.655               | 0.7375              | 0.685               |
> | Loss + **SDR**          | **0.767**              | **0.705**               | **0.796**           | **0.750**               | **0.834**              | **0.785**               |
> | Min-K             | 0.587              | 0.580               | 0.6767              | 0.670               | 0.7472              | 0.705               |
> | Min-K + **SDR**       | **0.750**              | **0.715**               | **0.749**              | **0.700**               | **0.820**              | **0.765**               |
> | Min-K++           | 0.475              | 0.535               | 0.4749              | 0.515               | 0.5863              | 0.605               |
> | Min-K++ + **SDR**     | **0.586**              | **0.615**               | **0.538**              | **0.580**               | **0.617**              | **0.615**              |
>
> Across all three mixed-register prompts, SDR consistently improves both AUC and ASR, demonstrating that it can successfully transfer to mixed laundering transformations.
>
> Nevertheless, we agree that extremely exotic transformations (e.g., pseudo‑translation into low-resource languages) may further stress SDR, and we will strengthen Appendix I to highlight that more task‑specific taxonomies are an important direction for future work rather than a solved problem.
>
>
> ---
> ##### References
> ###### [1] Wang, Xun, et al. "Pay attention to your tone: Introducing a new dataset for polite language rewrite." arXiv preprint arXiv:2212.10190 (2022).

---

> > ### Author Response · Authors · 2025-11-23
> > **Response with expanded ablation and comparative experiments**
> >
> > # Response with expanded ablation and comparative experiments
> >
> > We thank the reviewer for pointing out the need for more expanded ablation and comparative experiments. We address W2 and W3 below.
> >
> > > **W2.** SDR needs repeated calls to an auxiliary model to build templates and iterate prompts. The experiment section only includes a limited introduction of query budgets, latency, and sensitivity to n, m, l, and K. A more comprehensive sensitivity study would help strengthen the paper.
> >
> >
> > We have conducted an extended sensitivity study to examine how SDR behaves under different choices of the key hyperparameters $K$, $l$, $m$, and $n$, using Loss+SDR under the No.1 inside-register prompt.
> >
> > Our key findings are (see the tables below for detailed numbers):
> > - increasing $K$ from $3 \to 15$ improves AUC from 0.71 to 0.75 and ASR from 0.69 to 0.71, at the cost of higher runtime and query budget (roughly `$7` to `$39`with GPT-4o, substantially cheaper with GPT-4o-mini)
> > - value of $m$ has the strongest impact. Changing $m$ between $3 \to 9$ raises AUC from 0.72 to 0.81 and ASR from 0.69 to 0.76 with almost unchanged query cost, making $m = 9$ a reasonable default.
> > - $l$ and $n$ exhibit moderate improvements. Setting $l=7$ and $n=7$ balances performance and cost. *Note that $n$ is used only once when building templates and does not affect per‑run latency.*
> >
> >
> > #### Detailed numerical results on hyperparameter sensitivity
> > We use $K$=10, $l$=5, $m$=5, and $n$=10 as the default hyperparameters unless otherwise specified.
> > | $K$ | AUC   | ASR   | Time    | Query Budget (GPT4o/GPT4omini) |
> > |---|-------|-------|-----------|--------------|
> > | 3 | 0.712 | 0.685 | 02:23:42  | ~\$7/0.8         |
> > | 5 | 0.727 | 0.690 | 04:11:33  | ~\$13/1.5        |
> > | 15| 0.747 | 0.705 | 13:27:12  | ~\$39/4.5        |
> >
> > | $l$ | AUC   | ASR   | Time      | Query Budget (GPT4o/GPT4omini) |
> > |---|-------|-------|-----------|--------------|
> > | 3 | 0.715 | 0.670 | 08:21:33  | ~\$10/1        |
> > | 7 | 0.731 | 0.695 | 09:13:12  | ~\$10/1         |
> > | 9 | 0.718 | 0.695 | 09:35:17  | ~\$10/1         |
> >
> > | $m$ | AUC   | ASR   | Time      | Query Budget (GPT4o/GPT4omini) |
> > |---|-------|-------|-----------|--------------|
> > | 3 | 0.718 | 0.685 | 08:14:33  | ~\$10/1         |
> > | 7 | 0.796 | 0.740 | 08:19:29  | ~\$10/1         |
> > | 9 | 0.810 | 0.760 | 08:26:45  | ~\$10/1         |
> > | 11|0.806 | 0.755 | 08:33:34| ~\$10/1 |
> >
> > | $n$ | AUC   | ASR   | Time      | Query Budget (GPT4o/GPT4omini) |
> > |---|-------|-------|-----------|--------------|
> > | 3 | 0.705 | 0.675 | —         | —            |
> > | 7 | 0.752 | 0.705 | —         | —            |
> > | 9 | 0.742 | 0.695 | —         | —            |
> >
> > *Note: $n$ controls the construction of the template and is computed only once at initialization; therefore, it does not contribute to the runtime or query budget.*
> >
> > Overall, a practical and efficient configuration for SDR is $(K=5\, l=7\, m=9\, n=7)$, which attains most of the performance gains at moderate cost. We will add these tables and a short discussion to the Appendix.

---

> ### Author Response · Authors · 2025-11-23
>
> > **W3.** Results compare SDR plus standard detectors to the detectors alone. Since the contribution is a laundering-aware search, comparisons to other data-centric countermeasures or prompt search strategies would help to improve.
>
> We agree that positioning SDR relative to other search strategies is valuable.
> To the best of our knowledge, existing *reverse prompt engineering* methods, however, assume that the unknown prompt is applied at *inference* and leaves a direct trace in observed outputs. In our setting, the laundering prompt is applied before *training*. The data rights holder only sees a trained model and never observes any text generated under the unknown laundering prompt, making a direct application impossible.
>
> We summarize the differences between [2] and SDR below.
> | | Prompt Reverse Engineering[2] | Laundering Transformations Search |
> |---|---|---|
> | **Prompt applied** | During inference | Before training |
> | **Prompt visibility** | Prompt leaves a trace because it directly conditions model outputs.| Laundering goals leave only implicit effects in the trained model. |
> | **Access to outputs** | Has direct access to outputs generated under the prompt | Cannot access outputs generated under the laundering prompt; only the trained model of outputs generation is observable |
>
> To still provide a comparison, we slightly adapt [2] to a use case that better suits our problem setup: given an original document, we let $M_t$ generate a continuation and treat that continuation as a proxy "laundered" version; we then ask [2] to infer a prompt connecting the original and continuation.
>
> We obtain the results below, presenting a comparison between SDR and [2].
> | Method | AUC | ASR |
> |--------|-----|-----|
> | **Loss + SDR** | **0.782** | **0.764** |
> | Loss + Reverse Prompt Engineering [2] | 0.682 | 0.663 |
> | Loss | 0.638| 0.635 |
>
> We therefore conclude that, even in a favorable setting for [2], SDR provides substantially larger gains. We will include this comparison and a short conceptual discussion in the revised Appendix.
>
>
> ##### References
> ###### [2] Li, Hanqing, and Diego Klabjan. "Reverse prompt engineering." arXiv preprint arXiv:2411.06729 (2024).
>
> ---
> We sincerely hope these clarifications address the reviewer’s concerns. If there are any remaining questions, we would be happy to address them. Otherwise, we would appreciate if the reviewer could reconsider and raise the scores.

---

> ### Author Response · Authors · 2025-11-27
> **Follow-up on rebuttal clarifications**
>
> Dear Reviewer zZxJ,
>
> With the discussion period closing in less than 6 days, we are writing to kindly follow up on our rebuttal.
>
> We would greatly appreciate it if you could take a moment to review our response. If you find that these new results address your concerns, we would value the reconsideration of your assessment.
>
> We remain available to answer any further questions you might have.
>
> Best regards,
>
> The Authors

---

### Official Review · Reviewer_W3wo · 2025-11-01

**Soundness:** 2
**Presentation:** 1
**Contribution:** 2
**Rating:** 4
**Confidence:** 4

**Summary:**

This paper studies the task of data contamination (or membership inference) detection, particularly in scenarios where the training data have been laundered (e.g., undergone some register transfer) prior to model training. Previous work on data contamination or membership inference has primarily focused on detecting unauthorized data use on the exact same data, without considering potential laundering transformations.

As I understand it, the paper proposes to “reverse the laundering process”, i.e., first by identifying the most likely laundering register (i.e., the stylistic or structural form into which the original data may have been transformed), and then by recovering finer-grained data details through iterative prompting of an auxiliary LLM. Experimental results show that existing data contamination detection methods experience a substantial performance drop on laundered/synthetic samples, while the proposed reverse process helps recover several detection metrics.

**Strengths:**

- The paper investigates an important and interesting topic that has not been extensively explored in the existing literature.

- The proposed method demonstrates several intriguing empirical results.

**Weaknesses:**

- The threat model (and the corresponding protocol) is somewhat unclear. Several entities (such as the target model $M_t$, the auxiliary model $M_a$, and the datasets $D_{pro}$ and $D_{held}$) seem to have implicit assumptions, but these are not clearly stated. For example, is $D_{held}$ guaranteed to contain only non-member samples (i.e., data that were never used in training $M_t$)? Is the proprietary data $D_{pro}$ fully or partially assumed to have been used during training? These are important clarifications, especially for a work claiming to detect “unauthorized data usage”, which typically requires strict, verifiable integrity assumptions to be meaningful in practice. In addition, the data setup for evaluation remains somewhat vague (See detailed questions below).

- The method description is somewhat confusing. Stronger intuition and clearer formulation of the design ideas would help. For example, in Algorithm 2, the pseudo-code suggests that a single “system prompt” is always returned, whereas the text description implies that it may instead correspond to sample-level prompts. It might be clearer to use explicit sample indices or a more formalized notation to distinguish between them.

**Questions:**

- How are member versus non-member samples defined when computing AUC, ACC, or TPR in Tables 2–5? Are they randomly split from the same MIMIR dataset? If so, how do you ensure that the non-member data were indeed not used during the pretraining of the target model (given that many large models are trained on massive web-scale corpora that may overlap with MIMIR?) Some explicit discussion of data provenance control and efforts made to ensure disjoint membership would be very helpful. Also, what exactly does the detector observe, the laundered data samples, original samples, or both?

- Related to the above concern about the unclear threat model, it remains unclear what exactly is meant by “unauthorized training data detection on $M_t$” in the construction of $Perf_r$ (see Algorithm 1, line 20, and Algorithm 2, line 11). Why is it considered a reasonable setting to compute this directly on the target model that is itself under test? Would this not constitute a form of data leakage or evaluation exposure?  In this sense, a clear and explicit description of the different data subsets (e.g., the member data, non-member data, held-out data, and potentially shadow data) and their respective roles in training and evaluation is essential, but currently missing. This clarification is particularly critical for a submission claiming to address the detection of unauthorized training data.

- Algorithm 1, Line 22: variable naming inconsistency — $perf_r$ should be in uppercase ($Perf_r$) for consistency.

- Algorithm 1 explicitly specifies the auxiliary model as GPT-5, but Algorithm 2 does not mention which model is used.

---

> ### Author Response · Authors · 2025-11-23
> **Response for threat model clarification**
>
> # Response for threat model clarification
>
> We thank the reviewer for pointing out places where the threat model and notation were unclear. We address W1 and Q1-Q4 below. (original weaknesses and questions are quoted for reference)
>
> > W1. The threat model (and the corresponding protocol) is somewhat unclear. Several entities (such as the target model $M_t$, the auxiliary model $M_a$, and the datasets $D\_{\text{pro}}$ and $D\_{\text{hold}}$) seem to have implicit assumptions, but these are not clearly stated.
>
> Our setting follows a black-box auditing formulation used in post-hoc unauthorized data detection [1][2], describing a threat model where:
> - The model provider trains the target LLM $M_t$ on an unknown training set $D_{\rm train}$ that may include laundered variants of some samples in a proprietary dataset $D_{\rm pro}$.
> - The data rights holder owns a proprietary dataset $D_{\rm pro}$ and has a held-out reference set $D_{\rm held}$ guaranteed not to appear in $D_{\rm train}$. The data rights holder has only access to (i) $D_{\rm pro}$, (ii) $D_{\rm held}$, and (iii) black-box query access to $M_t$ (e.g., API). It has no access to $D_{\rm train}$, nor to the training and (potential) laundering pipeline.
> - An auxiliary LLM $M_a$ is any model the rights holder can query to synthesize candidate surrogate samples.
>
> We have also summarized a notation table for clarity.
>
> | Symbol| Description & role in the threat model | Visible to data‑rights holder? |
> |-|- | -|
> | $M_t$ | **Target LLM.** Trained by the model provider on $D_{\text{train}}$.| ✘ (black‑box only) |
> | $M_a$ | **Auxiliary LLM.** Used by data rights holder to synthesize candidate surrogates and SDR rewrites (*never* used for training $M_t$). | ✔|
> | $T$ | **True laundering transformation.** *Unknown* transformation applied by model provider to samples of $D_{\text{pro}}$ before training.| ✘ |
> | $p$  | **Reverse‑synthesis prompt.** Natural‑language specification inferred by SDR. Used to approximate $T$ when executed by $M_a$.  | ✔ |
> | $D_{\text{pro}}$ | **Proprietary dataset.** Original texts owned by the data‑rights holder and suspected of possible misuse. Used as the “candidate” set in detection.| ✔  |
> | $D_{\text{pro}}'$   | **Unknown in‑training subset of $D_{\text{pro}}$.** Elements of $D_{\text{pro}}$ whose laundered variants appear in $D_{\text{lau}}$.  | ✘ |
> | $D_{\text{train}}$  | **Model provider’s training dataset** for $M_t$. May contain laundered versions of some proprietary data plus arbitrary additional data.| ✘ |
> | $D_{\text{lau}} = T(D_{\text{pro}}')$ | **Laundered proprietary training data.** Obtained by applying $T$ to an unknown subset $D_{\text{pro}}' \subseteq D_{\text{pro}}$, i.e., training samples w.r.t. laundering.| ✘ |
> | $D_{\text{held}}$ | **Held‑out non‑training reference set.** Constructed to be disjoint from $D_{\text{train}}$ (e.g., post‑release data in MIMIR). Used as the reference corpus for detectors.| ✔|
> | $s$ | **Original sample.** Generic sample $s \in D_{\text{pro}} \cup D_{\text{held}}$. | ✔|
> | $\hat{s} = M_a(p, s)$| **Synthetic surrogate for $s$.** Rewrite of $s$ produced by $M_a$ under prompt $p$; SDR’s estimate of how $s$ might have been laundered.| ✔|
> | $\text{Syn}_p(S)$  | **Prompt‑induced surrogate set.** $\mathrm{Syn}\_p(S) = \{\, M_a(p, s) \mid s \in S \,\}$, where $S \subseteq D_\mathrm{pro} \cup D_\mathrm{held}$. Used as “training‑like” queries to $M_t$. | ✔ |
> | $\text{Perf}_p$ | **Detection performance under prompt $p$.** Scalar metric of $f$ when separating $\mathrm{Syn}\_p(D\_\mathrm{pro})$ from $\mathrm{Syn}\_p(D\_\mathrm{held})$ on $M_t$. SDR maximizes $\text{Perf}_p$. | ✔ |
>
>
> To assist in further clarifying the following questions, we first briefly revisit the **goal of SDR**: to test whether any laundered variant of samples in $D\_{\rm pro}$ appears in $D\_{\rm train}$, based on the data rights holder's query and data access permission.
>
> **What SDR actually optimizes.**
> SDR seeks to optimize a prompt $p$, such that the samples in $D_{\rm pro}$ can be recovered to the real laundered training data used to train $M_t$, so that existing unauthorized training data detection methods (e.g., ReCaLL, Min-K, etc.) can regain their effectiveness.
> 1. Uses $M_a$ to synthesize $\text{Syn}\_p(D\_\text{pro})$ and $\\text{Syn}\_p(D\_\text{held})$.
> 2. Runs an off‑the‑shelf detector (ReCaLL, Min‑K, etc.) on $M_t$ using these two sets, exactly as in standard post‑hoc unauthorized data detection but using the synthesized variants.
> 3. Takes the detector’s output scalar score (e.g., ReCaLL’s two‑sample score) as $\text{Perf}_p$, and SDR seeks to optimize a prompt $p$ that maximizes $\text{Perf}_p$.

---

> ### Author Response · Authors · 2025-11-23
>
> > **Q1.** For example, is the proprietary data $D_{pro}$ fully or partially assumed to have been used during training?
>
> In the threat model, $D_{\rm pro}$ is a *candidate pool*, from which an arbitrary, unknown subset of $D\_{\rm pro}$ may be laundered as $D\_{\rm lau}$ and included in $D\_{\rm train}$. Generally, SDR does *not* limit **how many samples within $D\_{\rm pro}$ are used** for laundering and training $M_t$.
>
> In our original paper, we instantiate this by training $M_t$ on the full use of $D_{\rm pro}$ and $D\_{\rm lau} = D\_{\rm train}$. Nevertheless, we acknowledge there are cases where the model provider may launder only a subset of $D_{\text{pro}}$ for training.
> Our problem setting does **not** assume full laundering. We further empirically validate that SDR remains feasible under **partial laundering**, and we report the corresponding results at:
> https://openreview.net/forum?id=GbNQyQEJOo&noteId=8iGUylnNKP.
>
> > **Q2.** it remains unclear what exactly is meant by “unauthorized training data detection on $M_t$ in the construction of ${\rm Pref_p}$ (see Algorithm 1, line 20, and Algorithm 2, line 11)?
>
> Given a fixed detector (e.g., ReCaLL, Min‑K), a prompt $p$ induces synthetic surrogates ${\rm Syn}\_p (D_{\rm pro})$ and ${\rm Syn}\_p (D_{\rm held})$ via $M_a$. We then run the detector on $M_t$ precisely as in prior work, but using these surrogates as candidate vs. reference sets, and record a scalar performance measure ${\rm Pref_p}$. Algorithms (1-2) describe our pipeline that searches over the prompt space to maximize this scalar.
>
> - ${\rm Pref_p}$ distinguishes ${\rm Syn}\_p (D\_{\rm pro})$ from ${\rm Syn}\_p (D\_{\rm held})$ implies that $M_t$ was indeed trained on laundered version of $D_{\rm pro}$ and $p$ can recover the unknown laundering process used by the model provider.
> - Otherwise, it implies that the current prompt $p$ cannot recover the laundered training data and we need to continue to search for more plausible ones.
> - Eventually, if no such performance-improving prompt can be found, we consider no laundering of $D_{\rm pro}$ was used in training $M_t$.
>
> We will revise Sec. 4 to provide an explicit and clear definition.
>
> > **Q3.** Why is it considered a reasonable setting to compute this directly on the target model that is itself under test? Would this not constitute a form of data leakage or evaluation exposure?
>
> We do **not** consider this to constitute data leakage.
> Under our threat model, the auditor is *by design* allowed to query $M_t$.
> SDR operates under the premise that *if and only if* the model provider has indeed laundered $D_{\rm pro}$, SDR would be able to improve the performance of off-the-shelf detectors.
> In this context, SDR is only tasked with searching for natural language prompts to adapt the inputs so that off-the-shelf detectors can **better distinguish** between ${\rm Syn}\_p (D_{\rm pro})$ and ${\rm Syn}\_p (D_{\rm held})$.
> We note that this pipeline differs fundamentally from training a machine learning model and thus does **not** constitute evaluation exposure.
>
> To address the concern of whether SDR triggers a false alarm (i.e., improve ${\rm Pref_p}$ even when the model provider did not launder $D_{\rm pro}$.), we add a **negative-control experiment** where neither $D_{\rm pro}$ nor its laundered variants were used.
> In this setup, SDR fails to find prompts that improve detection: across detectors, AUC and ASR remain close to 0.5 on both ArXiv and Wikipedia, indicating that SDR does *not* spuriously create evidence of misuse when $D_{\rm pro}$ was not used. We will include this negative-control analysis in the appendix.
>
> **Table: Negative-control experiments**
>
> | Method| ArXiv AUC | ArXiv ASR | Wiki AUC | Wiki ASR |
> |-|-|-|-|-|
> | RecaLL |0.516| 0.532 | 0.528| 0.545 |
> | RecaLL + **SDR**  | 0.488| 0.525 | 0.473 | 0.540 |
> | Loss | 0.486 | 0.522| 0.471| 0.510  |
> | Loss + **SDR**  | 0.490 | 0.535  |0.465| 0.515|
> | Min-K | 0.536 | 0.510 | 0.486|0.515 |
> | Min-K + **SDR** | 0.499 | 0.535| 0.440 | 0.525|
> | Min-K++ |0.476| 0.521| 0.483|0.535|
> | Min-K++ + **SDR**|0.483| 0.520| 0.446| 0.515|
>
> > **Q4.** A clear and explicit description of the different data subsets (e.g., the member data, non-member data, held-out data, and potentially shadow data) and their respective roles in training and evaluation is essential, but currently missing.
>
> We will revise Sec. 4 by adding a short subsection and supplementing a notation table (as with the beginning of this response) in the appendix, clearly defining the roles and data/model accessibility under our setup.
>
> ---
> ##### References
> ###### [1] Xie, Roy, et al. "ReCaLL: Membership Inference via Relative Conditional Log-Likelihoods." Proceedings of the 2024 Conference on Empirical Methods in Natural Language Processing.
>
> ###### [2] Choi, Hyeong Kyu, et al. "How Contaminated Is Your Benchmark? Measuring Dataset Leakage in Large Language Models with Kernel Divergence." Forty-second International Conference on Machine Learning.

---

> > ### Author Response · Authors · 2025-11-23
> > **Response for text and data clarifications**
> >
> > Response for text and data clarifications
> > ---
> > We thank the reviewer for identifying several issues and ambiguities in our writing and data descriptions. We address W2 and Q5-Q8 below. (original weaknesses and questions are quoted for reference)
> >
> > > **Q5.** How are member versus non-member samples defined when computing AUC, ACC, or TPR in Tables 2–5? Are they randomly split from the same MIMIR dataset? If so, how do you ensure that the non-member data were indeed not used during the pretraining of the target model (given that many large models are trained on massive web-scale corpora that may overlap with MIMIR?) Some explicit discussion of data provenance control and efforts made to ensure disjoint membership would be very helpful.
> >
> > We note that the MIMIR benchmark is specifically constructed for evaluating unauthorized training data detection. For each model to be evaluated (Pythia, Falcon, LLaMA-2), it defines "seen" splits (i.e., **ground-truth** training data) against "non-seen" splits (i.e., non-training data), which contain documents collected *after* the model's release and thus **not used in its pretraining** [3]. This therefore ensures the "non-seen" splits thus can be reliably used in our evaluation.
> >
> > Our experiments (Tables (2-5)) strictly rely on the MIMIR's construction: we use samples only from the non-seen split. From this pool we select 200 documents, and then fine-tune $M_t$ on a random half portion (with laundered variants, as $D_{\rm train}$) while holding out the other half (as $D_{\rm held}$).
> >
> > We will clarify this protocol and introduce MIMIR in the Appendix.
> >
> > > **Q6.** What exactly does the detector observe: the laundered data samples, the original samples, or both?
> >
> > The detector **never** observes the model provider's actual laundered training samples. Baseline detectos operate on $(D_{\rm pro}, D_{\rm held})$, and SDR-augmented detectors operate on the synthetic surrogates $({\rm Syn}\_p (D_{\rm pro}), {\rm Syn}\_p (D_{\rm held}))$ produced by $M_a$. The real laundered training set remains unknown.
> >
> > This matches the realistic auditing setting where the laundered data is processed by the model provider. **The laundered data, as well as the laundering pipeline, are undisclosed to the data rights holder.**
> >
> > *Please also refer to the beginning of the Response for threat model clarification.*
> >
> > > **W2.** The method description is somewhat confusing. Stronger intuition and clearer formulation of the design ideas would help. For example, in Algorithm 2, the pseudo-code suggests that a single “system prompt” is always returned, whereas the text description implies that it may instead correspond to sample-level prompts. It might be clearer to use explicit sample indices or a more formalized notation to distinguish between them.
> >
> > > **Q7.** Variable naming mismatch (e.g., $pref_r$ vs. $Pref_r$).
> > > **Q8**. The auxiliary model is explicitly stated in Algorithm 1 but not in Algorithm 2.
> >
> > We apologize for the ambiguity. **To clarify, SDR returns a single system-level prompt, as correctly reflected in Algorithm 2**. The per-sample expressions such as "Editing $p$ enables the transformation of $\hat{s}$ into $\tilde{s}$" are only intermediate descriptions passed to $M_a$, which then returns an updated global prompt $p$.
> >
> > We will revise Algorithm 2 to explicitly list $M_a$ in the inputs, and add sample indices $s_i, \hat{s}_i, \tilde{s}_i$ to distinguish system-level from sample-level objects, and standardize notation (i.e., always using ${\rm Pref}_r$). We expect these edits to remove the remaining confusion about what the algorithm returns.
> >
> > ##### References
> > ###### [3] Chunyuan Deng, Yilun Zhao, Xiangru Tang, Mark Gerstein, and Arman Cohan. Investigating data contamination in modern benchmarks for large language models. arXiv preprint arXiv:2311.09783, 2023.
> >
> >
> > ---
> > We sincerely hope these clarifications address the reviewer’s concerns. If there are any remaining questions, we would be happy to address them. Otherwise, we would appreciate if the reviewer could reconsider and raise the scores.

---

> ### Author Response · Authors · 2025-11-27
> **Follow-up on rebuttal clarifications**
>
> Dear Reviewer W3wo,
>
> With the discussion period closing in less than 6 days, we are writing to kindly follow up on our rebuttal.
>
> We would greatly appreciate it if you could take a moment to review our response. If you find that these new results address your concerns, we would value the reconsideration of your assessment.
>
> We remain available to answer any further questions you might have.
>
> Best regards,
>
> The Authors

---

> ### Author Response · Authors · 2025-12-02
> **Supplementary Experiments Addressing Reviewer W3wo Q1**
>
> ### Supplementary Experiments Addressing Reviewer `W3wo` Q1
> We conducted an experiment to evaluate whether SDR remains effective when only a portion of the proprietary data is laundered. Given $|D_{\text{pro}}|$ = 200, we randomly selected half of the samples and applied the first five inside-register prompts (Appendix Table 9) to generate the laundered subsets used for training, while the remaining samples were left unaltered. SDR was then applied to the full $D_{\text{pro}}$ to infer the laundering transformation. The table shown below reports the average performance across all five prompts.
>
> ### Average Performance Across All Prompts
>
> | Method            | Avg AUC | Avg ASR |
> |-------------------|---------|---------|
> | RecaLL            | 0.608 | 0.590 |
> | RecaLL + SDR      | **0.722** | **0.694** |
> | Loss                | 0.599 | 0.597 |
> | Loss + SDR          | **0.732** | **0.703** |
> | Min-K             | 0.608 | 0.603 |
> | Min-K + SDR       | **0.724** | **0.701** |
> | Min-K++           | 0.553 | 0.565 |
> | Min-K++ + SDR     | **0.580** | **0.602** |
>
> SDR successfully improves detection performance across all four off-the-shelf detectors.
> For example, Loss’s AUC/ASR increases from 0.599/0.597 to 0.732/0.703.

---

### Meta-Review · Area_Chair_zXXv · 2025-12-30

**Summary:**

The paper evaluates detecting training data in LLMs when this data was „laundered“ before, this means transformed by a model owner who does not want the legitimate data owners to find out that they trained on their data. During the rebuttal, the authors ran large amounts of experiments to address the major **empirical concerns** of the reviewers, including 1) the generalization of the method to other re-writing methods than the ones tested, including human-based rewriting, 2) an analysis of the compute overhead for data laundering, 3) comparison against other baselines, and 4) extension to mixed registers. Additionally, the authors clarified the threat space and the intuition of the method.

**Reviewer Concerns:**

After a careful review by the AC, the following **conceptual** concerns from the reviewers and AC remain:
1. A major concern is that membership inference attacks were chosen for the setup, even though, the community has come to the conclusion that „Membership inference attacks cannot prove that a model was trained on your data“ [3]. In fact, it has been shown that the weak (i.e., non-shadow model-based) MIAs evaluated in this work do not outperform random guessing when preventing distribution shift between member and non-member data [1]. In fact, it was even theoretically proven that, as training data grows, MIA performance at random guessing [4]. As a solution, dataset inference for LLMs [2] was introduced. This might be a more solid departure ground for evaluation in the paper.
2. The distribution shift is a concern that the paper should address and investigate: do all experiments in the paper (despite having „random“ rewrites) really achieve the same distribution? Or is it possible that the training members (one rewrite) just differ in distribution from the chosen non-members. Getting this point very sound, e.g. with a baseline model evaluated on both distributions (before tuning), as done in prior work, would be helpful to make this point.
3. The experimental setup is also not yet clear enough: One major concern raised by Reviewer 1 (R1) is: how do we know that the results are correct, given that when we operate on a pretrained model, we don’t know if it already had our chosen non-member data in training (making that data member data without us noticing). This setup will need to be clarified. For Pythia, where the training data is public, the AC sees a chance for the authors to ensure not accidentally blurring results, due to Pythia training data being online available. For models like Llama, where it is not, this is hard.
4. While the rebuttal now includes an evaluation of the cost overhead of doing the laundering (and it seems so small that one can argue it is cheaper for a model owner than paying for the data), it is unclear, what training on laundered data does to the model performance. It was shown that models can collapse when trained on synthetic data (rather than real data [5]), yet, the paper only report the attack success, but not the model performance (e.g. train vs test loss). Doing so will have two benefits: 1) It will help the authors analyze overfitting (which might cause the high membership risks observed), and 2) assess whether the launderings really worth it for the model owner, in the sense of not only being cheap enough, but also yielding equally good models than training on the original data.
Given these points, the AC is sure that the paper will make a valuable contribution to a future conference, but has too many open directions to explore for acceptance to ICLR.

**References**

[1] Das, Debeshee, Jie Zhang, and Florian Trantèr. "Blind baselines beat membership inference attacks for foundation models." In 2025 IEEE Security and Privacy Workshops (SPW), pp. 118-125. IEEE, 2025.

[2] Maini, Pratyush, Hengrui Jia, Nicolas Papernot, and Adam Dziedzic. "LLM Dataset Inference: Did you train on my dataset?." Advances in Neural Information Processing Systems 37 (2024): 124069-124092.

[3] Zhang, Jie, Debeshee Das, Gautam Kamath, and Florian Tramèr. "Position: Membership Inference Attacks Cannot Prove That a Model was Trained on Your Data." In 2025 IEEE Conference on Secure and Trustworthy Machine Learning (SaTML), pp. 333-345. IEEE, 2025.

[4] Maini, Pratyush, Mohammad Yaghini, and Nicolas Papernot. "Dataset inference: Ownership resolution in machine learning." arXiv preprint arXiv:2104.10706 (2021).

[5] Shumailov, Ilia, Zakhar Shumaylov, Yiren Zhao, Nicolas Papernot, Ross Anderson, and Yarin Gal. "AI models collapse when trained on recursively generated data." Nature 631, no. 8022 (2024): 755-759.

**Reviewer Scores:**

The discussion period would have likely not sufficed to get the paper to an acceptance.

---

### Decision · Program_Chairs · 2026-01-26

Reject